# A unified framework for estimating country-specific cumulative incidence for 18 diseases stratified by polygenic risk

Bradley Jermy [1,25], Kristi Läll[2,25], Brooke N. Wolford [3,25], Ying Wang [4,5], Kristina Zguro[6], Yipeng Cheng[7], Masahiro Kanai [4,5], Stavroula Kanoni [8], Zhiyu Yang[1], Tuomo Hartonen [1], Remo Monti [9], Julian Wanner [1,9], Omar Youssef [10,11], Estonian Biobank research team*, FinnGen*, Christoph Lippert [9,12], David van Heel [13], Yukinori Okada [14,15,16,17], Daniel L. McCartney [7], Caroline Hayward [18], Riccardo E. Marioni [7], Simone Furini[6,19], Alessandra Renieri [6,20,21], Alicia R. Martin [4,5], Benjamin M. Neale [4,5], Kristian Hveem[3,22], Reedik Mägi [2], Aarno Palotie [1,4,5], Henrike Heyne [1,9,12], Nina Mars [1,4,5], Andrea Ganna[1,23,26] ✉ & Samuli Ripatti [1,23,24,26] ✉

Polygenic scores (PGSs) offer the ability to predict genetic risk for complex diseases across the life course; a key benefit over short-term prediction models. To produce risk estimates relevant to clinical and public health decision-making, it is important to account for varying effects due to age and sex. Here, we develop a novel framework to estimate country-, age-, and sex-specific estimates of cumulative incidence stratified by PGS for 18 high-burden diseases. We integrate PGS associations from seven studies in four countries ($N = 1,197,129$) with disease incidences from the Global Burden of Disease. PGS has a significant sex-specific effect for asthma, hip osteoarthritis, gout, coronary heart disease and type 2 diabetes (T2D), with all but T2D exhibiting a larger effect in men. PGS has a larger effect in younger individuals for 13 diseases, with effects decreasing linearly with age. We show for breast cancer that, relative to individuals in the bottom 20% of polygenic risk, the top 5% attain an absolute risk for screening eligibility 16.3 years earlier. Our framework increases the generalizability of results from biobank studies and the accuracy of absolute risk estimates by appropriately accounting for age- and sex-specific PGS effects. Our results highlight the potential of PGS as a screening tool which may assist in the early prevention of common diseases.

Clinical calculators are often used to estimate disease risk in common diseases to facilitate early identification and primordial prevention. Clinical calculators are designed from clinical and epidemiological knowledge of disease risk factors and validated in the best set of representative data available. Designed to be generalizable, clinical calculators give physicians a simplified framework for clinical decision-making, with risk thresholds often dictating whether patients should receive earlier or more frequent screening in the case of breast cancer

A full list of affiliations appears at the end of the paper. *Lists of authors and their affiliations appear at the end of the paper.
✉e-mail: andrea.ganna@helsinki.fi; samuli.ripatti@helsinki.fi

or pharmaceutical and behavioral interventions in the case of cardio-vascular disease. QDiabetes[1] (https://qdiabetes.org/), and the pooled cohort equations estimate risk for type 2 diabetes (T2D) and athero-sclerotic cardiovascular disease[2], respectively using sex, age, smoking status, clinical information, and biomarkers such as cholesterol. The breast and ovarian analysis of disease incidence and carrier estimation algorithm v6 now includes monogenic (e.g., BRCA) and polygenic risk factors[3].

Polygenic scores (PGSs) use combined information from a person's genome to estimate their genetic risk of developing a specific disease or trait[4]. Most of the predictive ability of PGS is obtained by summing thousands of common genetic variants of small effect, but PGS can also incorporate rare genetic variants with large effect[5]. There is extensive discussion about the clinical and public health value of PGSs[6–8] with varying values on short-term prediction when integrated on top of existing clinical prediction models for cardiovascular diseases and prostate cancer[9–11]. Other authors have highlighted how PGSs provide independent, and therefore complementary, information about disease risk compared to many key risk factors included in prediction models such as family history[7,12]. PGSs have been shown to associate strongly with many diseases and stratify individuals based on their genetic risks[13,14]. Further, their impact on global disease burden, as measured by disability-adjusted life years (DALYs), is comparable to well-established modifiable risk factors[15].

Maybe the most attractive feature of PGSs is that they can be calculated at birth, allowing for risk estimation in younger individuals not typically targeted by current disease risk calculators[16,17]. The static nature of PGS means risk can be computed over the lifetime using a single genetic test for many diseases simultaneously, making them a potentially cost-effective prediction tool[6]. On the contrary, current clinical calculators provide the absolute risk of disease over a short time frame, typically the next 5 years or 10 years and are only applicable within a limited age range[1,2], which may partially explain why such calculators typically fail to identify high-risk individuals with early-onset disease causing the biggest burden to the society and for the affected individual[18].

Thus, PGSs are a potentially useful tool in overcoming the limitations of short-term risk prediction and are well suited to provide lifetime absolute risk estimates. Such estimates must be comprehensively reviewed if they are to be used in personalized screening approaches. First, given countries may differ in terms of disease incidence and the discriminative ability of PGS, thus, we must understand how PGS generalizes across countries and health systems. A recent international study examining PGS association with 14 diseases across seven countries only focused on relative risk[19]. Second, while there is some evidence of a larger genetic contribution to early-onset disease cases[20–22], a detailed understanding of how risk estimation of PGS varies by both age and sex and how this translates to estimates of cumulative incidence is required to improve accuracy. Third, most Biobank studies are not representative of the general population[23], with only a few studies having attempted to recalibrate the impact of PGS on disease prevalence[24]. We address these three questions as part of the INTERnational consortium of integratiVE geNomics prEdiction (INTERVENE, https://www.interveneproject.eu/).

In this work, we introduce a novel framework to allow for country-specific stratification opportunities for risk-based prevention and screening strategies. We demonstrate our method by combining incidences with polygenic risk associations across seven studies in four countries ($N = 1{,}197{,}129$) for 18 high-burden diseases[25]. We demonstrate that for many diseases, PGSs stratify individuals into distinct risk trajectories over the lifetime with large differences in cumulative incidence between groups. Our results also show that in many diseases the PGS effects are sex- and age-specific. To put our results into context and demonstrate the potential translational utility of our approach, we provide examples of how these results can be used for improving risk-based disease screening in different countries.

## Results

### Descriptive statistics

Contributing Biobank size ranged from 7018 participants (after filtering to GP and hospital data consented individuals) in GS to 447,332 participants in UKB. GE had the youngest median age of recruitment at 26.3 years (28 interquartile range (IQR)) and GS had the oldest median age of recruitment at 57 years (17.6 IQR, Table 1). FinnGen had the longest follow-up with 62 years (19 IQR) and MGB had the shortest follow-up with 10 years (13 IQR). EstBB had the largest percentage of female participants (66%) and HUNT had the least (53%).

Of the 18 phenotypes of interest, cancer was generally the most common phenotype (prevalence range 9–47%) except for EstBB which had a 26% prevalence of depression and a 9% prevalence of cancer. The least common phenotype was type 1 diabetes (T1D) in EstBB, MGB, and UKB, melanoma in FinnGen and HUNT, and rheumatoid arthritis in GS and GE, all at less than 1% prevalence (Supplementary Data 1). Across phenotype and Biobanks, the oldest median age of onset was prostate cancer in EstBB and GS, lung cancer in FinnGen, and MGB, atrial fibrillation in GE and HUNT, and gout in UKB (Supplementary Data 1). Appendicitis had the earliest age of onset for all Biobanks (range = 23.7–55.7 years) except for FinnGen and UKB, which had the youngest age of onset for T1D at 12.9 years (19.2 IQR) and 55.7 years (16.13 IQR), respectively.

### Association between PGS and 18 diseases

Due to insufficient sample size, poor phenotype definition within a specific study, or non-independent PGS not every trait was analyzed

**Table 1 | Descriptive statistics by study**

| Study | Sample size | Age of recruitment (yrs) median (IQR) | Maximum follow-up time across traits (yrs) median (IQR) | % Female | Ascertainment strategy |
|---|---|---|---|---|---|
| Estonia Biobank | 199,868 | 43.5 (25.5) | 17.7 (0) | 65.5 | Population |
| FinnGen | 412,090 | 55.8 (26.7) | 62 (19) | 55.9 | Population and hospital |
| Genomics England | 29,427 | 26.3 (28) | 29 (0) | 59 | Hospital |
| Generation Scotland | 7018 | 57 (17.6) | 39.8 (0) | 56.4 | Population |
| HUNT | 69,715 | 37 (21.1) | 25 (18) | 52.9 | Population |
| Mass General Brigham Biobank | 39,036 | 51 (22) | 10 (13) | 55.1 | Hospital |
| UK Biobank | 447,332 | 58 (12) | 24 (0) | 54.2 | Population |

Follow-up time for some studies is defined as the start of registry data until the end of the last linking between the registry and Biobank (see Supplementary Methods). For example, Estonian Biobank uses as a baseline the start of National Health Insurance Fund data from 2003 so the follow-up IQR is 0. Survival models used age as a timescale defining follow-up as age from birth.

*yrs* years, *IQR* interquartile range.

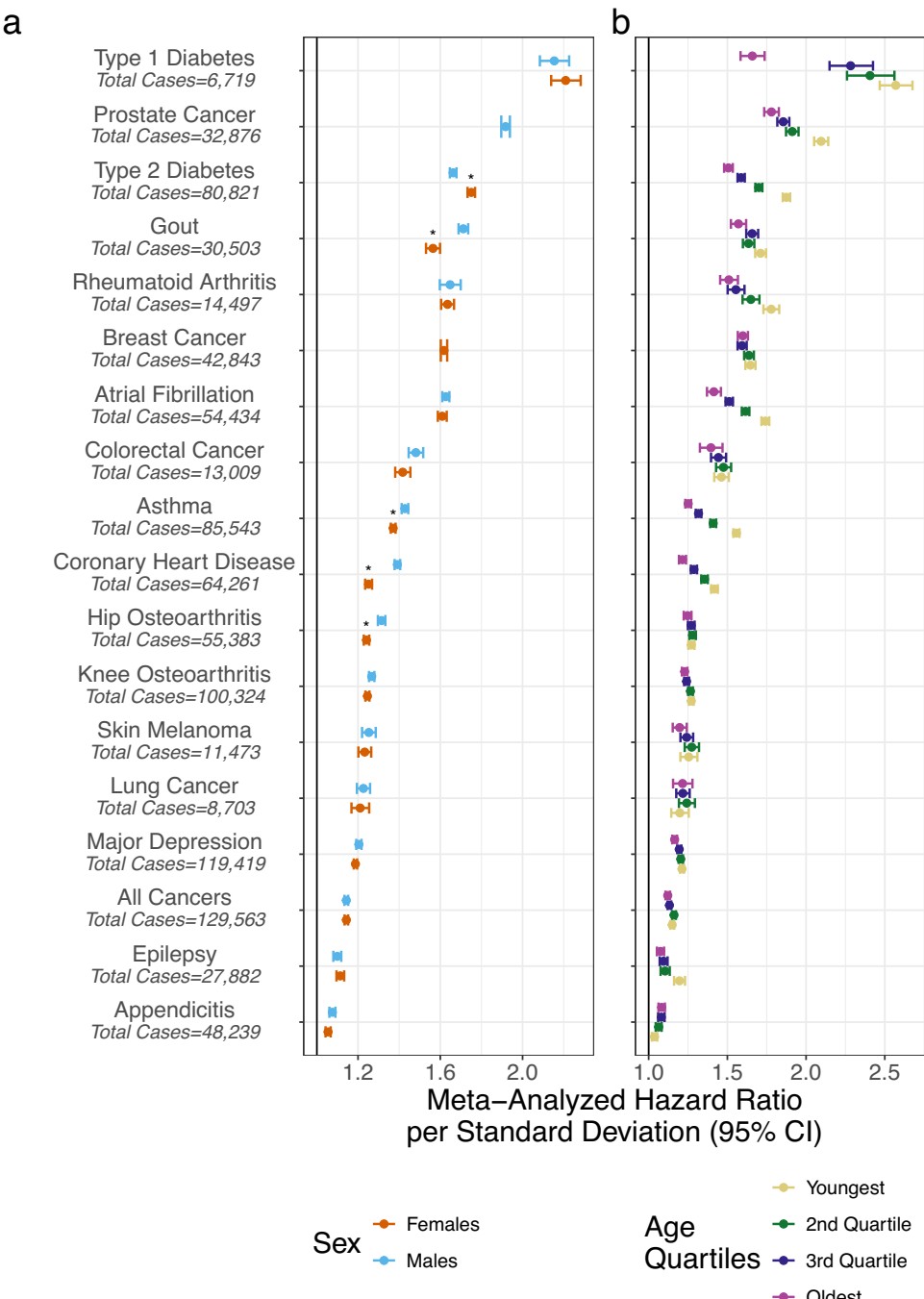

**Fig. 1 | Model selection for each phenotype. a** Data are presented as the fixed-effects meta-analysis of the log hazard ratios per standard deviation of PGS stratified by sex. **b** Meta-analysed hazard ratios per standard deviation stratified by age. The asterisk indicates a significant interaction between PGS and sex estimated by a two-sided Wald test after Bonferroni correction for multiple testing of 18 phenotypes ($p$ value $< 2.8 \times 10^{-3}$). The exact $p$ values are in Supplementary Table 1. Case and control sample sizes for each phenotype in each biobank are in Supplementary Data 1.

within each biobank. 18 diseases were analyzed in 3 biobanks (EstBB, FinnGen and Genomics England), 15–17 of them in three biobanks (HUNT, Generation Scotland, and Mass General Brigham) and six diseases in UKB. All PGSs were significantly associated with 18 respective diseases with an HR for 1 standard deviation in the PGS ranging from 1.06 (95% CI: 1.05–1.07) for appendicitis to 2.18 (95% CI: 2.13–2.23) for T1D. We observed significant heterogeneity, as tested by Cochran's Q test, in estimates of relative risk across studies, partially driven by the large sample size which allowed us to detect small, yet significant differences (Supplementary Fig. 1 and Supplementary Data 2). Two examples of large study heterogeneity were observed for coronary

heart disease (CHD), which PGS had HRs per SD ranging from 1.13 (95% CI: 1.07–1.19) in GS to 1.41 (95% CI: 1.40–1.43) in FinnGen and T1D with HRs per SD ranging from 1.41 (95% CI: 1.17–1.69) in MGB to 2.37 (95% CI: 2.31–2.44) in FinnGen.

### Sex- and age-specific effects
We identified significant interactions between disease-specific PGS and sex for five diseases ($p < 2.8 \times 10^{-3}$; Supplementary Table 1). PGS had a larger effect on CHD, gout, hip osteoarthritis, and asthma in men whereas for T2D the effect was larger in women (Fig. 1a and Supplementary Data 2). The change in the PGS effect with age was particularly

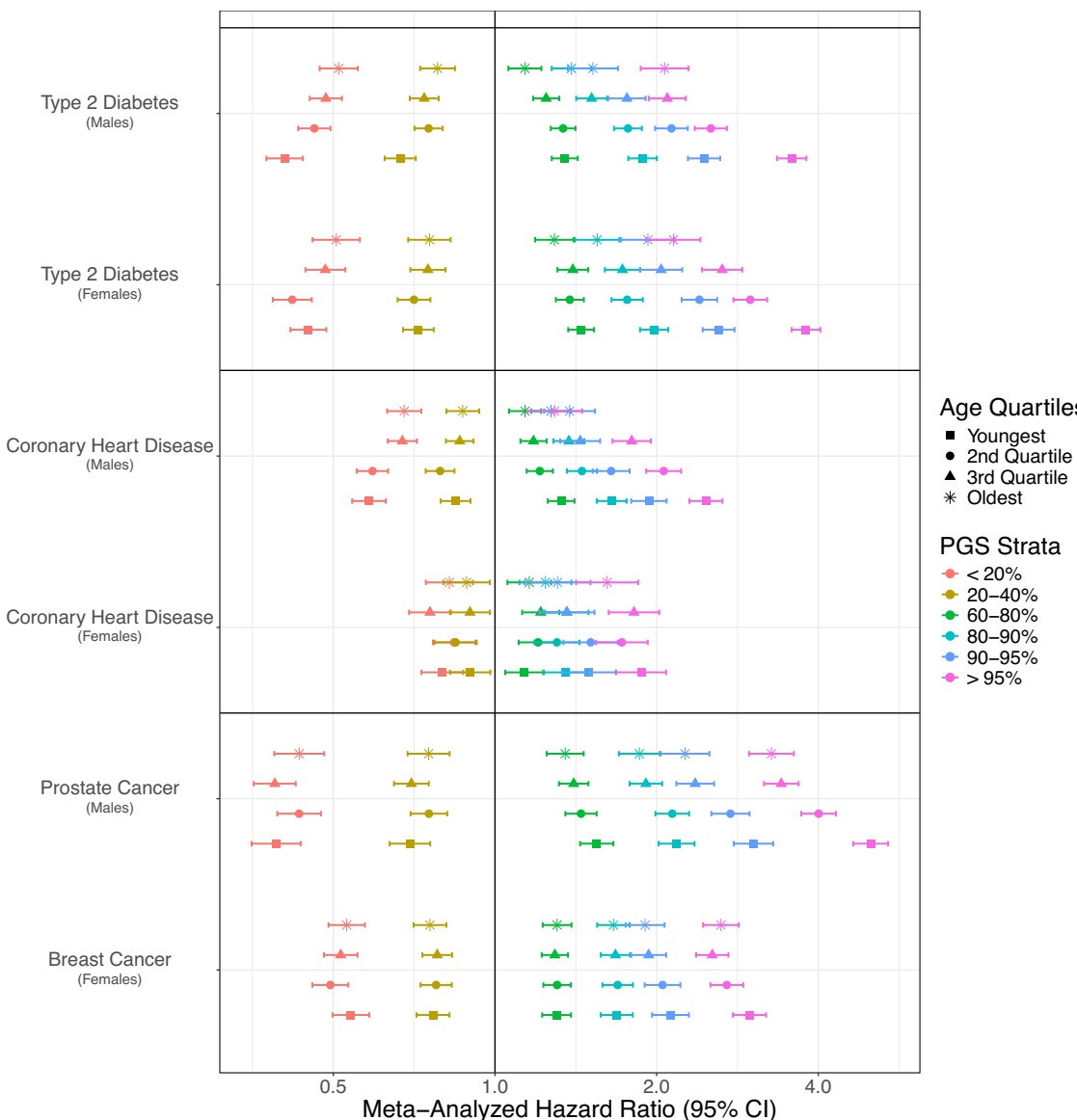

**Fig. 2 | Meta-analyzed hazard ratios stratified by age and sex.** Data are presented as the fixed-effects meta-analysis of the log hazard ratios per standard deviation of PGS stratified by sex, age quartile, and PGS strata for T2D, CHD, prostate cancer, and breast cancer. Case and control sample sizes for each phenotype in each biobank are in Supplementary Data 1.

prominent. In total, significant heterogeneity across age quartiles was detected in 13 of 18 phenotypes (Cochran's Q $p$ value $< 2.8 \times 10^{-3}$ for all cancers, appendicitis, asthma, atrial fibrillation, CHD, epilepsy, gout, depression, knee osteoarthritis, prostate cancer, rheumatoid arthritis, T1D and T2D) (Fig. 1b and Supplementary Data 3). The decreasing effect of PGS was approximately linear with age (Supplementary Fig. 2) and relatively consistent across studies (Supplementary Fig. 1c). The differences in age effects were large for T1D where the PGS effect per standard deviation was 2.57 (95% CI: 2.47–2.68) in the youngest quartile (age < 12.6) and 1.66 (95% CI: 1.58–1.74) in the oldest quartile (age > 33.3).

The large sample size allowed us to further examine the combined effect of both age and sex on PGS associations. One notable example was the association between PGS and CHD which decreased with age only in men, but not in women ($P_{het}$ in men = $1.05 \times 10^{-44}$; $P_{het}$ in women = 0.04) (Supplementary Figs. 3 and 4; Supplementary Data 2; and Supplementary Table 2).

Further examining the association between PGS and 18 diseases by PGS quantiles (Fig. 2) we identified that for some diseases, age-specific effects were larger among individuals belonging to the tails of the PGS distribution. For example, individuals in the top 5% of a PGS for prostate cancer vs those in the 40–60% reference group had a significantly higher relative risk for prostate when the disease was diagnosed at a younger age (age < 62.6) HR = 5.01 (95% CI: 4.65–5.39) compared to oldest ages (age > 73.9) HR = 3.27 (95% CI: 2.97–3.60). We also provide Harrell's C-statistic to compare models with sex- and age-specific effects to a baseline model (Supplementary Fig. 5).

**Country-specific cumulative incidence estimation stratified by PGS**

For each disease, we derived country-specific estimates of the cumulative incidence by PGS quantiles, accounting for age and sex-specific effects and calibrating the baseline risk using GBD. Supplementary Table 3 highlights the final models to be employed in the estimation of cumulative incidence.

Variation in cumulative incidence was evident by PGS quantiles, country and sex with the main driver of the difference between country and sex being the difference in baseline disease risk (Fig. 3;

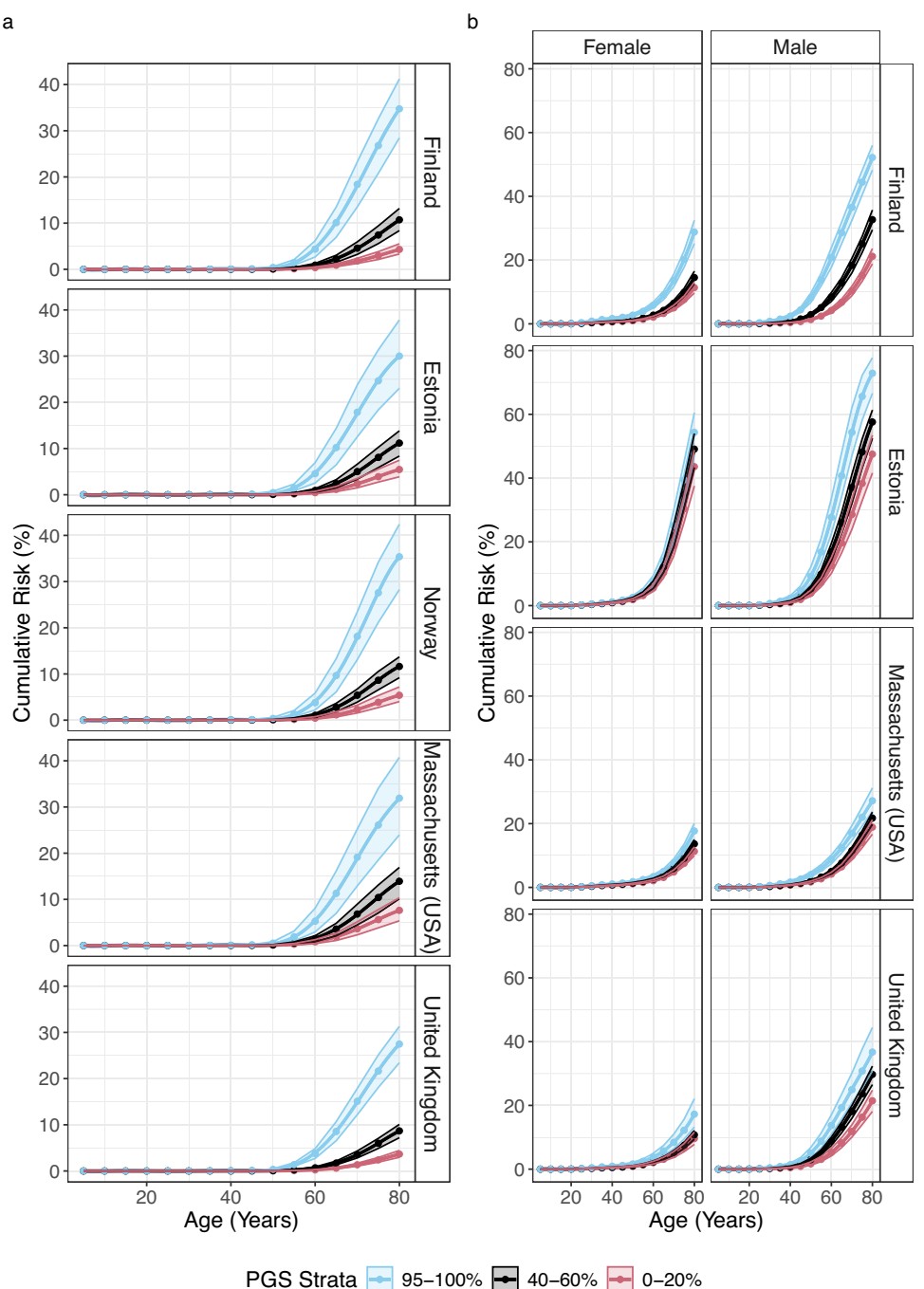

**Fig. 3 | Country and sex-specific cumulative incidence estimates.** Bootstrapped 95% confidence intervals reflect the uncertainty of the cumulative incidence estimates for the top, median, and bottom of the PGS distribution for **a** prostate cancer and **b** CHD.

Supplementary Fig. 6; and Supplementary Data 4). For example, in Massachusetts, the cumulative incidence at age 80 for CHD was significantly greater in men in the top 5% of PGS compared to the bottom 20% quantile (27.2 [95% CI: 23.8–31.2%] vs 18.9% [95% CI: 16.7–21.2%]). In Estonia, absolute differences between the top 5% and bottom 20% PGS quantiles were larger (72.9 [95% CI: 66.6–77.8%] vs 47.5% [95% CI: 41.6–53.4%]) due to overall higher incidence. In women, the absolute difference for the same PGS quantiles was lower than in men due to a decreased baseline risk (Massachusetts: 17.7% [95% CI: 15.3–20.2%] vs 11.3% [95% CI: 10.0–12.7%]); Estonia: 54.3% [95% CI: 47.2–60.5%] vs 43.6% [95% CI: 37.5–48.8%]). Similarly, for prostate cancer in the UK, the cumulative incidence was greater in men in the top 5% of PGS compared to the

bottom 20% quantile (27.5% [95% CI: 23.4–31.3%] vs 3.7% [95% CI: 3.0–4.4%]). In Norway, the cumulative incidence was greater for the same PGS quantiles (35.4% [95% CI: 28.3– 42.4%] vs 5.4% [95% CI: 4.0–7.2%]).

### PGSs and disease screening

Our GBD-calibrated country-specific cumulative incidence estimates allowed us to illustrate the potential utility of risk-based stratified screening for two diseases with existing screening recommendations: T2D and breast cancer. As mentioned earlier we found that the tails of the PGS distribution are particularly impacted by age-specific effects (Fig. 2). This has direct relevance for screening strategies given clinical decisions regarding treatment will more likely occur in these groups.

We, therefore, further explored the improvement in calibration that results from accounting for age- and sex-specific effects.

The American Diabetes Association recommends universal screening for T2D at age 45 with 3-year check-ups if the results of the screening are normal[26,27]. We therefore estimated the country-specific cumulative incidence at age 45 prior to PGS stratification and used this as a clinical threshold (Supplementary Fig. 7). Thresholds in cumulative incidence at age 45 varied substantially across countries; ranging from 5.6% (Estonia) to 13.0% (UK) in men and from 4.7% (Norway) to 8.9% (UK) in women (Supplementary Table 4).

We estimated at which age the risk thresholds would have been reached as a function of PGS. Across studies, individuals in the bottom 20% of PGS would reach the risk threshold at an average age of 63.1 (95% CI: 58.8–67.3) whereas individuals in the top 5% would reach the same risk threshold at an average of 29.3 (95% CI: 26.2–32.9); a difference of 33.8 years (Supplementary Figs. 6p and 8 and Supplementary Data 5). If age- and sex-specific PGS effects were not accounted for in the calculation of the cumulative incidence, the ages at which the risk threshold was attained would be, on average, 2.8 years earlier in the bottom 20% and 1.8 years later in the top 5%.

For breast cancer, in many countries, the initial screening is recommended to women at age 50[28–33]. In the countries examined by our study, the average cumulative incidence at age 50 ranged from 1.47% (Norway) to 2.05% (UK) (Supplementary Fig. 9 and Supplementary Table 5). Women in the bottom 20% of PGS reached the risk threshold for breast cancer screening, on average, at age 58.7 (95% CI: 55.8–62.3) whereas women in the top 5% reached it at an average age of 42.9 (95% CI: 41.6–44.9); a mean difference of 15.8 years (Supplementary Figs. 6e and 10 and Supplementary Data 6).

To further illustrate the effects of the cumulative incidence differences at the tails of the PRS on potential risk-based screening, we used the largest biobank study FinnGen to estimate the top/-bottom 1% PRS cumulative incidences and calculated the ages at which the cumulative incidences were at the same level as in average person in the Finnish population at the start of the ADA screening for T2D (45 years) and national screening program for breast cancer (50 years). For T2D, Men and women in the top 1% reached the threshold aged 24.8 (95% CI: 22.5–27.6) and 22.3 (95% CI: 20.0–25.3), respectively. Individuals in the bottom 1% of PGS did not reach the risk threshold by age 80 (Fig. 4a). For breast cancer women in the top 1% of PGS reached the threshold aged 42.3 (95% CI: 41–44.3) whereas women in the bottom 1% of PGS reached the threshold aged 66.3 (95% CI: 61.3–72.4); a difference of 24 years (Fig. 4b).

### Translating to additional countries

In practice, most countries are unlikely to have studies with sufficient power to obtain robust associations between PGS and diseases. A possible solution is to use a pooled estimate from the meta-analysis across studies, however, heterogeneity in HR across countries could limit the tools' utility. Despite such heterogeneity, it appears that meta-analyzed HRs are a good substitute. Using T2D and CHD as our examples, we first created a pooled estimate of the HRs by meta-analyzing all estimates and recalculating cumulative incidence by combining these HRs with country-specific baseline hazards. In general, country-specific cumulative incidences using the meta-analyzed HRs were within the confidence intervals of our original estimates of cumulative incidence (Supplementary Data 7); indicating meta-analyzed HRs are a good substitute in the absence of country-specific data. Where differences did exist cumulative incidence was elevated when using meta-analyzed HRs. For T2D, cumulative incidence was increased in the tails of the PGS distribution (top 5% and bottom 20%) in Massachusetts, as well as for Norwegians in the bottom 20% only (Supplementary Fig. 11a). Similarly for CHD, cumulative incidence was elevated in Estonian women and men from

Massachusetts in the top 5% of polygenic risk (Supplementary Fig. 11b).

### Sensitivity analyses

We performed sensitivity analyses in the six phenotypes considered in the UKB—gout, rheumatoid arthritis, prostate cancer, breast cancer, T1D, and epilepsy. First, we determined that the inclusion of related individuals did not impact the association between PGSs and diseases (Supplementary Fig. 12). Second, we evaluated the robustness of our disease definitions based on primary care data from the UKB. In the non-cancer phenotypes tested in the UKB, disease definitions using primary care data resulted in reduced HRs relative to the secondary care phenotypes. However, this difference was removed when adding in the criterion for each individual to have at least two codes (Supplementary Fig. 13)—a common practice within primary care phenotyping to reduce misclassification[34]. Combining primary and secondary care data tended to produce an association closer to the primary care-only phenotype. We did not test breast and prostate cancer because these are generally well-tagged by hospital diagnoses and cancer registries. Third, survival bias, potentially induced by considering cases before study enrollment, does not seem to impact our results. Among six phenotypes only prostate cancer PGS had a reduced relative risk when considering follow-up at baseline—equivalent of testing incident cases only—rather than birth (Supplementary Fig. 14 and Supplementary Table 6).

## Discussion

In this study, we use data on 1.2 million study participants from seven biobank studies to provide a broad overview of the impact of PGS on the cumulative incidence of 18 diseases. We find evidence of considerable heterogeneity in the effect of PGS by both age and sex. We integrate such variation to reflect more accurate estimations of cumulative risk over the life course and highlight how PGS stratifies individuals and can impact risk-based screening practices for breast cancer and T2D.

Our findings allow us to draw several conclusions. First, the heterogeneity in PGS effects across ages shows that while our genetic profiles and PGS do not change with age, their impact on disease risk changes with age. A decreasing effect of PGS with age has been shown previously for some diseases[21,22,35] and our results confirm and expand those findings. In some diseases, environmental effects become more prevalent and variable with age, potentially from the accumulation of gene by environment interactions over the life course[22], in effect reducing the heritability which represents the upper bound of prediction from PGS. Our findings mirror those found in high-susceptibility genes, for example, *BRCA1* and *BRCA2* mutations have been found to be associated with an earlier age at onset for breast cancer[36]. Failing to account for the age-specific effects of PGS would underestimate disease risk in younger individuals.

Second, disease incidence is known to vary with sex for many diseases and this is mirrored by the sex differences in PGS effects[37–40]. While there is some evidence of sex-specific effects at the genetic variant level[41–43], it is limited, possibly owing to the greater power requirements for an interaction effect. Recent work suggests there are sex differences in the magnitude of genetic effects rather than in the actual causal variants[44], but there is much to consider when testing models of sex differences[45,46]. When combining thousands of genetic variants in a PGS, we show significant sex-specific associations for five diseases. Among these diseases, CHD and T2D have some previous evidence to support sex-specific PGS associations[47,48]. There are many possible explanations for this differential effect. Different biological causal mechanisms could exist by sex. For example, lipids known to influence the risk for CHD and T2D have been shown to vary by sex and age[49]. Furthermore, the GWAS used to estimate the PGS may have an imbalance in the male:female ratio or sex-differential participation bias

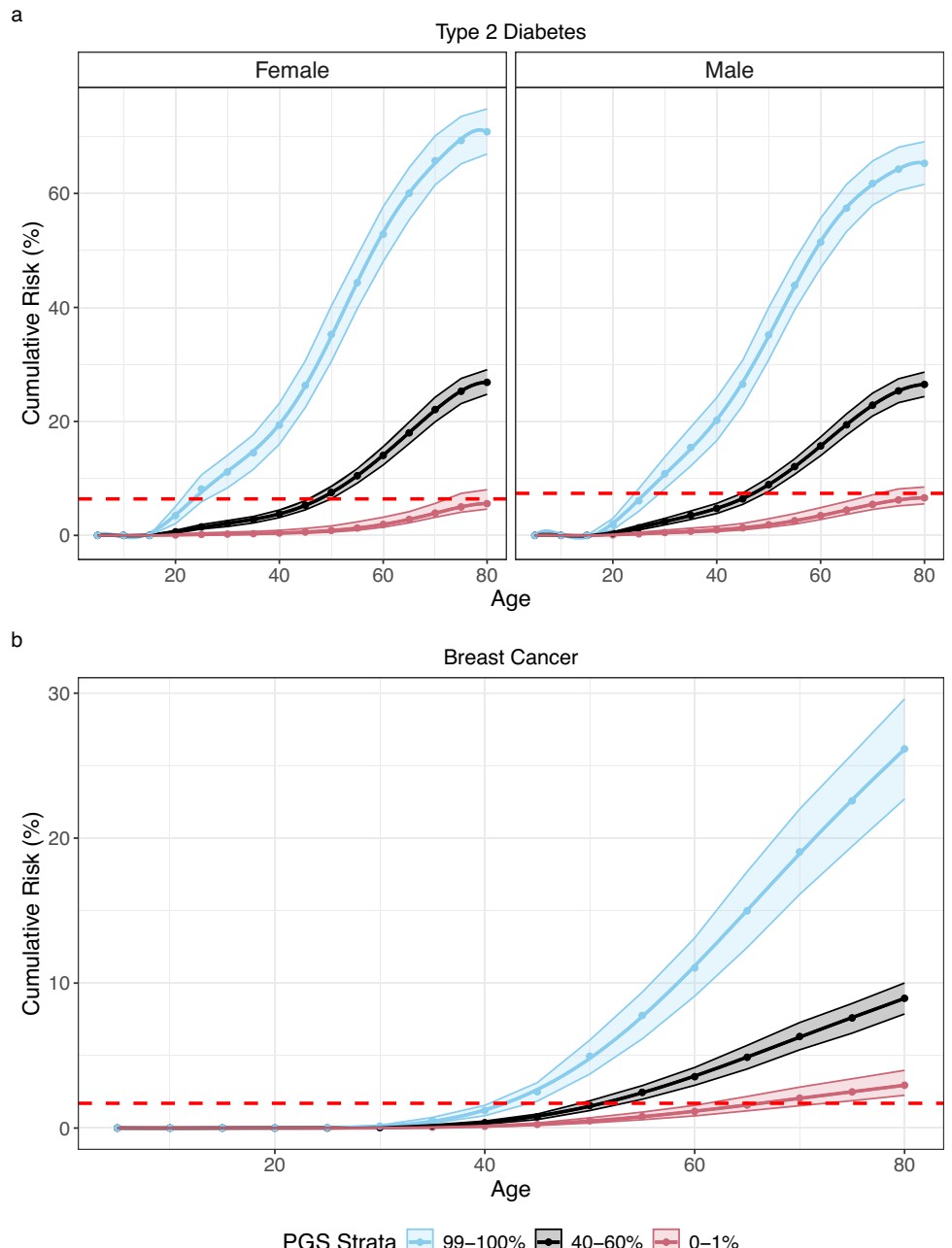

**Fig. 4 | Sex-specific cumulative incidence estimates for T2D and breast cancer in Finland.** The red dashed line in each figure represents a country-specific clinically defined cumulative incidence risk threshold for screening. Bootstrapped 95% confidence intervals reflect the uncertainty of the cumulative incidence estimates for the top, median, and bottom of the PGS distribution for **a** T2D and **b** breast cancer.

can induce differences in the phenotypic composition of males and females recruited in the study[50]. Sex-specific effects in the GWAS used for PGS construction could result in a stronger PGS association in the more common sex. For Gout and T2D men were 60% and 51.8% of the GWAS sample, respectively. While Gout showed a more significant PGS effect in males, T2D had a more significant effect in females. Finally, diagnostic bias may mean one sex needs to have a higher disease liability before receiving a diagnosis. Understanding the root cause of these differences is important and can ultimately only be solved through the routine introduction of sex-specific GWAS or GWAS in age and sex-stratified groups.

Third, we found that while PGS tends to have a larger effect on diseases at younger ages, this effect might differ between males and females. For CHD, we observed a large PGS-age interaction effect only

in males. Age at onset in CHD varies more in males than in females[25,51] and the risk factor profiles are known to change differently over age in males and females. Our results show that cumulative genetic effects captured by PGS are contributing to differences in age-related risks and may also be partly mediated by changes in risk factor profiles. In addition to CHD, we saw similar effects for gout which also has an earlier age at onset for males[52,53]. Even when accounting for age and sex (Supplementary Fig. 1), we see heterogeneity of PGS effects across biobanks, which could point to gene-by-environment interactions. Ultimately, each disease will require its own assessment for age- and sex-specific effects. This study provides such a method for selecting the PGS estimate which balances power and accuracy in a systematic fashion, while also accounting for the importance of a harmonized phenotype across studies.

Arguably the best application of our country-specific disease incidence and PGS estimates across large-scale biobanks is the development of country-specific risk calculators that are based on the lifetime risk of diseases which may be used to determine optimal ages for screening. Our framework is flexible in that it can integrate both country-specific and pooled PGS associations, depending on the requirement for specificity or generalizability, respectively. This approach has multiple important advantages relative to the short-term prediction of most current models. Lifetime risk can be estimated early in life and overcomes the well-documented challenge of short-term risk calculators for diseases not being useful in early life and therefore unable to enable a primordial prevention approach[18]. As we illustrate with T2D and Breast Cancer, PGS provides lifetime risk trajectories that can enable stratified screening approaches. Modeling age and sex-specific effects will be important for accurate identification of the age to begin screening. Further research is necessary to understand the optimal way to account for the uncertainty of the risk estimates for individuals in the tails of the PGS distribution, particularly when invasive intervention rather than increased screening is required. When available, other risk factors can easily be added when designing prediction models and screening approaches. Our approach also allows for developing country-specific risk calculators by utilizing baseline risk estimates derived from GBD, producing more accurate estimates for screening and overcoming the ascertainment biases inherent in many biobanks. We here demonstrate the risk estimation framework using 18 diseases, but risks can in principle be calculated for hundreds of diseases over the life course at little additional expense[6].

Our study should be considered in light of the following limitations. First, the PGS used in this study does not consider the risk due to rare genetic variation[54]. While this may reduce the accuracy of our risk estimates, at a population level, common genetic variation will be more predictive of the variation in complex diseases which we are particularly well-powered to test in this study. As more sequencing studies become available, rare variants will also be included in the genome-wide risk estimation[5]. Second, the use of harmonized phenotypes can ignore the country-specific nuances of ICD coding due to billing strategies. We deliberately focused on harmonization of the analysis to reduce bias due to technical variation, a rare feature in projects involving multiple studies. Third, our PGS relative risks are applicable to individuals of European ancestry only. As more studies with non-European individuals become available, similar estimates can be derived for a range of ancestries and admixtures. In particular, risk estimates for an individual should be derived using a PGS matching their ancestry as closely as possible. This study used multi-ancestry GWAS for PGS construction when possible, but large GWAS today are from the majority of European samples with limited generalizability to non-European populations[55]. Furthermore, the baseline risk estimated from GBD is currently ancestry-agnostic, and ideally, population calibration could be done at a higher resolution to estimate an ancestry-specific baseline within a country. This is particularly important for recent immigrants, who may have a baseline risk more representative of their home country as opposed to the average risk in their current country of residence. Fourth, GWAS summary statistics depend on the makeup of the GWAS cohorts and are influenced by the proportion of females, advanced age cases, and clinical vs population-based studies. Fifth, some of the studies included are non-representative of the population due to sample recruitment strategies (volunteer- or hospital-based). Older cohorts may show fewer individuals at the high-risk tail of the PGS distribution due to early mortality. While this may bias the cumulative risk estimation, a key strength of this study is the use of GBD to reduce the impact of selection bias in the baseline hazard which will bring our estimates closer to the true cumulative incidence. Finally, when modeling age-specific risk we used the hazard ratio estimated for the closest age quartile for ages less than the first age quartile and greater than the fourth age quartile (see "Methods").

Through this approach, we did not estimate exact hazard ratios (HRs) outside of the age ranges for which we have data and statistical power to do so, but our assumption of constant hazards outside the age quartiles would, for example, estimate a 20-year-old carrying the same T2D risk as a 45-year-old (Supplementary Fig. 15). In a translational context, this could create underestimation or overestimation of risk leading to over or under treatment, unnecessary expenditures, missed diagnoses, etc. In the future with larger data sets, HR estimates on a higher resolution age grouping, such as deciles, would allow for more accurate estimates.

In conclusion, we demonstrate the heterogeneity in polygenic score estimates between males and females and across the lifespan in many diseases. While accounting for this heterogeneity, we developed a uniform framework to allow for the estimation of lifetime risk of diseases, stratified by individual's genetic profiles and provide country-specific estimates. This information, which is already available for major modifiable risk factors[56,57], but was not yet comprehensively available for genetic scores, will allow health policymakers to better design screening tools with the goal of assisting in the early prevention of common diseases.

## Methods

### Participating in studies in INTERVENE

Data from approximately 1.2 million participants of European ancestries were used across seven studies—UK Biobank (UKB)[58], FinnGen[59], Estonian Biobank (EstBB), Trøndelag Health Study (HUNT)[60–62], Generation Scotland (GS)[63], Genomics England (GE)[64], and Mass General Brigham Biobank (MGB)[65]. Each contributing study performed genotyping, imputation, variant quality control and ancestry assignment using their own methodology (Supplementary Methods).

### Disease selection

Diseases were selected according to their global burden as defined by DALYs from the Global Burden of Disease (GBD) 2019[25], and the availability of GWAS summary statistics for the creation of PGS. Using these considerations, we selected 18 diseases contributing 17.87% of total global DALYs (Supplementary Table 7). These diseases contribute to 25.02% of total DALYs in high socio-demographic index countries, of which all studies included in this analysis are based.

### Phenotype harmonization

To harmonize disease phenotypes across studies, we used definitions curated by a team of clinical experts in FinnGen[59] (Supplementary Data 8). The presence of any ICD-9 and ICD-10 codes included within the FinnGen phenotype was used to define cases across the remaining studies. Controls were defined as individuals without the relevant ICD codes for the disease.

All data used to define disease phenotypes was registry-based. Missingness within registry data may result from either incomplete follow-up or a diagnosis being received in a health care system not included within the registry data, i.e., primary care. For these reasons, it is difficult to quantify the missingness of registry data, however, a comprehensive overview of each study's registry information is provided in the Supplementary Methods.

A key step in the estimation of cumulative incidence is in calculating the baseline hazard which requires reference statistics from a nationally representative sample. We used incidence, prevalence and mortality estimates from the GBD for this step. To quantify the degree of overlap between the phenotypes defined from GBD and our disease definitions, and therefore justify the baseline hazard for each disease, we computed the percentage overlap of ICD codes across the two definitions. The number of records for each ICD code was extracted from the UKB data showcase (https://biobank.ndph.ox.ac.uk/showcase/). Overlap was high, with 10 diseases having a 100% overlap, and 14 having above 95% overlap (Supplementary Data 9). We originally considered diseases

with 70% phenotype overlap which included interstitial lung disease (ILD). However, the GBD baseline hazard estimates from GBD were highly heterogeneous and unrealistic for a generally rare disease. As such, it was decided that all phenotypes were suitable for baseline hazard estimation using GBD. While the overlap in ICD codes for major depressive disorder was 100%, further inspection suggests a liberal definition is used by the GBD where individuals only need to have suffered either of the two cardinal symptoms of MDD (depressed mood or anhedonia) over a two week period[25]. To reflect the fact this is not a clinical diagnosis, we use the term depression throughout the rest of the paper.

## Estimating PGSs

For each phenotype, we searched for the summary statistics from the GWAS with the greatest sample size that was publicly available within the GWAS catalog (Supplementary Data 10). Biobank and trait combinations were only studied if independent from the GWAS contributing studies. To enhance analytical consistency and ensure variants were of high quality, we used single nucleotide polymorphisms (SNPs) in the intersection of HapMap phase three SNPs and 1000 genomes[66–68] with a minor allele frequency greater than 1% in at least one superpopulation ($M = 1,330,820$). MegaPRS—a collection of PGS tools that allow the expected heritability contributed by each SNP to vary—was chosen for SNP weight calculation as a previous methods comparison paper has shown it to have equal or superior prediction across a range of phenotypes[69,70] (opain.github.io/GenoPred). We selected the BLD-LDAK heritability model as this is recommended by the authors and used a data-driven approach to tool and hyperparameter selection (we allow the data to find the best tool/hyperparameters by specifying the 'mega' argument). Following weight calculation, PLINK was used to generate PGS for participants in each study. PGS in analyses were standardized to have mean 0 and variance 1 for each study.

## Survival analysis models

We performed ancestry-specific Cox proportional hazards (PH) regression with age at disease onset as the timescale in each study. Follow-up starts at birth and ends at the age of the first record of a disease diagnosis (for individuals with the diseases), age at death for a cause other than the disease, age at last record available in the registries or electronic health records or age 80, whatever happened first. If the study was included in the base GWAS used for PGS calculation, the model was not tested with the exception of FinnGen and EstBB where relevant cohorts were excluded to remove sample overlap (Supplementary Data 10). In addition to the standardized PGS, the first ten genetic principal components and study-specific covariates used to control for technical artifacts (i.e., genotype batch, assessment center) were used as covariates.

## Sex and age stratification

Four separate Cox-PH models were tested for each phenotype: (1) using the full sample (no stratification), (2) sex-stratified, (3) age-stratified, and (4) age and sex-stratified). We first computed HRs per standard deviation within each study. We then performed a fixed-effects meta-analysis on the log HRs across all studies tested to understand the generalizability of any age- or sex-specific effects. This was performed with the metafor package in R[71]. Studies were only included within the meta-analysis if it was possible to estimate a log HR in every stratum, i.e. all age or sex strata.

To test for sex differences, an interaction term of PGS with sex was added to the model. In addition, Cox-PH models were repeated in each sex and HRs were compared for any significant differences (Supplementary Methods).

To test for age-specific effects, we stratified each disease into four intervals, calculated according to the mean age at onset quartiles

across FinnGen, HUNT, UKB, and EstBB (Supplementary Data 11). We then performed separate Cox models in each interval using a method previously described[22,72,73]. Briefly, disease onset was only considered within the interval of a given quartile and participants were considered censored at the end of the interval if they had not died of a separate cause. If the participant had the disease in a prior interval, they were excluded from any follow-up intervals.

When deciding upon the optimal model:

a. Sex-specific effects were chosen if:
  1. The inverse variance-weighted meta-analyzed interaction effect (PGS × sex) was significant
  ($p < 2.8 \times 10^{-3}$, details below)
b. Age-specific effects were chosen if:
  1. There was significant heterogeneity across the four quartiles, estimated using a Cochran's $Q$ test[74].
c. Age- and sex-specific effects were chosen if:
  1. Separate age and sex-specific effects were found in both tests a and b.
  2. Age-specific effects were found in a single-sex where not previously found.
  3. Age-specific effects were found to differ significantly across sexes.

To test if the age-specific effects were significantly different between men and women, we compared the effect sizes (Beta$_{men}$ = Beta$_{women}$, $p$ value < 0.05) for the weighted linear regression fit for log (HR) on age (Supplementary Methods).

In all instances, we use $p < 2.8 \times 10^{-3}$ as our significance threshold, which represents a Bonferroni correction of 18 tests (number of phenotypes). This is with the exception of testing age-specific effects across sexes where a nominal $p$ value threshold is used ($p < 0.05$) due to the limited data available between the age quartiles and HR.

## Cumulative incidence estimation

To calculate cumulative incidence—defined as the cumulative probability of disease from birth up to age 80 accounting for the competing risk of death from other causes—country and sex-specific estimates of age-specific (5-year age groups) incidence, prevalence and mortality were extracted for each disease from the GBD 2019[25].

Cumulative incidence was estimated using the method described[52]. Briefly, for each sex and age group, 5-year bins in the case of GBD, the disease incidence hazard for age group $[m, m + 5)$ was calculated as:

$$\text{hazard}_{[m,m+5)} = \text{incidence}_{[m,m+5)}/(1 - \text{prevalence}_{[m,m+5)}) \quad (1)$$

where incidence and prevalence values represent the number of new cases per year and the point prevalence assigned for the specific age group $[m, m + 5)$, $m = \{0, 5, 10, 15,...75\}$. The hazard for the age group therefore remains constant for all values within a given age group.

The hazard, in conjunction with the mortality rate due to other causes (overall mortality–cause-specific mortality), was then used to calculate the probability of survival up to age $k$, $k = \{0, 5, 10, 15,...80\}$:

$$\text{survival}_k = e^{-5*\sum_{m = \{0,5,10,...,75\},m<k}^{k} (\text{mortality}_{[m,m+5)} + \text{hazard}_{[m,m+5)})} \quad (2)$$

In Eq. 2, $m$ increments in steps of 5 to correspond to the age groups specified above. The combined mortality and hazard were multiplied by 5 to account for the fact that the hazard and mortality reported values per year yet the age group covers 5 years. Survival is equal to 1 at age 0.

Similarly, the probability of experiencing a given disease during age period $[m, m + 5)$ was calculated as:

$$\text{risk}_k = 1 - e^{-5 * \text{hazard}_{(m,m+5)}} \tag{3}$$

In Eq. 3, subscript $k$ corresponds to the lower bound of the age group $[m, m + 5)$ and $\text{risk}_k$ shows a 5-year risk for a person with age $m$. Cumulative incidence was then calculated as the cumulative sum of survival at a given age multiplied by the probability of the disease at that age. For example, lifetime cumulative incidence is the cumulative sum until age 80:

$$\text{Lifetime cumulative incidence} = \sum_{k=0}^{80} \text{survival}_k * \text{risk}_k \tag{4}$$

Cumulative incidence was calculated separately for each country (Estonia, UK, United States of America (USA), Norway and Finland) to ensure each study was calibrated to its population. As GE, GS and the UKB are all based in the UK, if association testing had been computed in more than one study, the HRs were meta-analyzed prior to computing cumulative incidence. As it was possible to estimate cumulative incidence at the state level, we calibrated the estimation to Massachusetts for MGB instead of using aggregated statistics for the USA.

To calculate cumulative incidence by PGS strata, it is necessary to group individuals according to their position in the PGS distribution. The default grouping was based on Mavadat et al.: <20%, 20–40%, 40–60% (reference), 60–80%, 80–90%, 90–95%, >95%. PGS We had higher resolution at the top end of the PGS distribution where screening has the most potential (e.g., >99%), but we were often underpowered to have symmetric resolution at the bottom end of the distribution (e.g., <1%) when stratifying by age and sex. Using these groupings, Cox-PH models were repeated in the full sample, and the sex, age, age and sex stratifications for each study.

Cumulative incidence for a given PGS group was calculated using the incidence estimates for the total population taken from the GBD and the HR from the relevant Cox-PH models. This method has been previously described[15]. Briefly, the total incidence for a given age is equal to the weighted average of incidences for each PGS group. Reference population incidence (incidence rates for PGS group 40–60%) was computed as:

$$I_0 = \frac{nI}{n_0 + \sum HR_i * n_i} \tag{5}$$

Where $I_0$ is the reference population, $I$ is the age-specific total incidence across the population, $n$ is the population size at a given age interval, $n_0$ is the population size of the reference population (40th–60th), $n_i$ is the population size for the i-th PGS group and $HR_i$ is the HR for the i-th PGS group.

The incidence attributable to an i-th PGS group was then estimated by:

$$I_i = I_0 * HR_i \tag{6}$$

Incidences were then converted to probabilities of experiencing the disease using Eqs. 1 and 3. Cumulative incidence for each PGS group was calculated using Eqs. 1 through 4 with mortality due to other causes assumed to be equal across PGS groups.

To quantify the degree of uncertainty in the cumulative incidence estimation, we randomly sampled from the distribution of the error for each estimated parameter for both baseline and Cox models and recalculated the cumulative incidence 1000×. 95% confidence intervals were calculated by taking the 2.5 and 97.5 percentiles of the results.

To incorporate age-specific effects into the estimation of cumulative incidence, it was necessary to estimate HRs across the age span (0–80). For each disease with age-specific effects, a weighted linear regression was fit to the log HR estimates from the four age quartiles with each estimate placed at the median age at onset in each quartile (Supplementary Fig. 15). Predicted HRs from this regression were then incorporated into the cumulative incidence estimation. Ages outside of the range of the four HRs were assumed constant to the HR closest in age (Supplementary Fig. 15).

## Translating estimates of cumulative incidence to other countries

To evaluate the impact of HR heterogeneity on the estimates of cumulative incidence, we meta-analyzed HRs for each PGS percentile group and re-evaluated cumulative incidence in each country using the country-specific baseline hazard. We then compared the cumulative incidence estimates for the study-specific HRs to the meta-analyzed HRs.

## Sensitivity analyses

**The impact of relatedness on PGS association—full cohort vs unrelated subset.** For EstBB and FinnGen, Plink v2[72] and KING[75] were used to identify all related individuals up to degree 3, respectively. Individuals were then excluded as to remove relatedness from the sample while retaining the maximal sample size. HRs resulting from the full sample were then compared to the unrelated subset.

**Robustness of registry-based disease definitions—primary care vs secondary care.** Many diseases are first diagnosed in a primary care setting. To review the impact of not including this data, we created four disease definitions using Read v2 and CTV3 codes from the primary care data in the UKB (Supplementary Data 12). To increase the consistency of our definitions to prior research, we used phenotype definitions from prior studies within the HDR UK Phenotype Library (https://phenotypes.healthdatagateway.org/). HRs calculated using the full sample were then compared to associations from our main definitions which used secondary care data only.

**The impact of the start of follow-up.** To evaluate the impact of using age as the timescale in the analysis, we tested two different follow-up times within the UKB. Firstly, we tested using age at recruitment as the start of follow-up, equivalent to only including incident cases. Secondly, we tested age at registry linkage. This was calculated by assuming the birth country remained the country at which the registry was first linked. If the individual was born in Wales the date of registry linkage was 1st January 1998, in Scotland the date was set as 1st January 1981 and if born anywhere else, the date was set as 1st January 1997 (the date in which English registries were linked). All other aspects of the main analysis were the same with the exception that year of birth was added as a covariate.

## Ethics statement

Patients and control subjects in FinnGen provided informed consent for biobank research, based on the Finnish Biobank Act. Alternatively, separate research cohorts, collected prior to the Finnish Biobank Act came into effect (in September 2013) and the start of FinnGen (August 2017), were collected based on study-specific consents and later transferred to the Finnish Biobanks after approval by Fimea (Finnish Medicines Agency), the National Supervisory Authority for Welfare and Health. Recruitment protocols followed the biobank protocols approved by Fimea. The Coordinating Ethics Committee of the Hospital District of Helsinki and Uusimaa (HUS) statement number for the FinnGen study is Nr HUS/990/2017.

The FinnGen study is approved by Finnish Institute for Health and Welfare (permit numbers: THL/2031/6.02.00/2017, THL/1101/5.05.00/2017, THL/341/6.02.00/2018, THL/2222/6.02.00/2018, THL/283/

6.02.00/2019, THL/1721/5.05.00/2019 and THL/1524/5.05.00/2020), digital and population data service agency (permit numbers: VRK43431/2017-3, VRK/6909/2018-3, VRK/4415/2019-3), the Social Insurance Institution (permit numbers: KELA 58/522/2017, KELA 131/522/2018, KELA 70/522/2019, KELA 98/522/2019, KELA 134/522/2019, KELA 138/522/2019, KELA 2/522/2020, KELA 16/522/2020), Findata permit numbers THL/2364/14.02/2020, THL/4055/14.06.00/2020, THL/3433/14.06.00/2020, THL/4432/14.06/2020, THL/5189/14.06/2020, THL/5894/14.06.00/2020, THL/6619/14.06.00/2020, THL/209/14.06.00/2021, THL/688/14.06.00/2021, THL/1284/14.06.00/2021, THL/1965/14.06.00/2021, THL/5546/14.02.00/2020, THL/2658/14.06.00/2021, THL/4235/14.06.00/2021, Statistics Finland (permit numbers: TK-53-1041-17 and TK/143/07.03.00/2020 (earlier TK-53-90-20) TK/1735/07.03.00/2021, TK/3112/07.03.00/2021) and Finnish Registry for Kidney Diseases permission/extract from the meeting minutes on 4th July 2019.

The Biobank Access Decisions for FinnGen samples and data utilized in FinnGen Data Freeze 10 include: THL Biobank BB2017_55, BB2017_111, BB2018_19, BB_2018_34, BB_2018_67, BB2018_71, BB2019_7, BB2019_8, BB2019_26, BB2020_1, BB2021_65, Finnish Red Cross Blood Service Biobank 7.12.2017, Helsinki Biobank HUS/359/2017, HUS/248/2020, HUS/150/2022 § 12, §13, §14, §15, §16, §17, §18, and §23, Auria Biobank AB17-5154 and amendment #1 (August 17 2020) and amendments BB_2021-0140, BB_2021-0156 (August 26 2021, Feb 2 2022), BB_2021-0169, BB_2021-0179, BB_2021-0161, AB20-5926 and amendment #1 (April 23 2020)and it´s modification (Sep 22 2021), Biobank Borealis of Northern Finland_2017_1013, 2021_5010, 2021_5018, 2021_5015, 2021_5023, 2021_5017, 2022_6001, Biobank of Eastern Finland 1186/2018 and amendment 22 § /2020, 53§/2021, 13§/2022, 14§/2022, 15§/2022, Finnish Clinical Biobank Tampere MH0004 and amendments (21.02.2020 and 06.10.2020), §8/2021, §9/2022, §10/2022, §12/2022, §20/2022, §21/2022, §22/2022, §23/2022, Central Finland Biobank 1-2017, and Terveystalo Biobank STB 2018001 and amendment 25th Aug 2020, Finnish Hematological Registry and Clinical Biobank decision 18th June 2021, Arctic Biobank P0844: ARC_2021_1001.

Ethics approval for the UK Biobank study was obtained from the North West Centre for Research Ethics Committee (11/NW/0382). UK Biobank data used in this study were obtained under approved application 78537.

The genotyping in Trøndelag Health Study and work presented here was approved by the Regional Committee for Ethics in Medical Research, Central Norway (2014/144, 2018/1622, and 2018/411492). All participants signed informed consent for participation and the use of data in research.

Ethical approval for the GS:SFHS study was obtained from the Tayside Committee on Medical Research Ethics (on behalf of the National Health Service).

The activities of the EstBB are regulated by the Human Genes Research Act, which was adopted in 2000 specifically for the operations of the EstBB. Individual level data analysis in the EstBB was carried out under ethical approval 1.1-12/624 from the Estonian Committee on Bioethics and Human Research (Estonian Ministry of Social Affairs), using data according to release application S22, document number 6-7/GI/16259 from the EstBB.

The informed consent process for the GE 100,000 Genomes Project has been approved by the National Research Ethics Service Research Ethics Committee for East of England−Cambridge South Research Ethics Committee.

The analysis using Mass General Brigham Biobank is approved under IRB protocol 2022P001736.

### Reporting summary

Further information on research design is available in the Nature Portfolio Reporting Summary linked to this article.

## Data availability

The raw individual-level data are protected and are not available due to data privacy laws, but they can be accessed through individual participating biobanks. The FinnGen data may be accessed through Finnish Biobanks' FinBB portal (www.finbb.fi; email: info.fingenious@finbb.fi). The Trøndelag Health Study (HUNT) may be accessed by application to the HUNT Research Centre at https://www.ntnu.edu/hunt/data. Researchers interested in EstBB can request access at https://www.geenivaramu.ee/en/access-biobank. De-identified data of the MGB Biobank that supports this study is available from the MGB Biobank portal at https://www.massgeneralbrigham.org/en/research-and-innovation/participate-in-research/biobank/for-researchers. Restrictions apply to the availability of these data, which are available to MGB-affiliated researchers via a formal application. UK Biobank data are available through a procedure described at http://www.ukbiobank.ac.uk/using-the-resource/.Genomics England data is available through an application process described here: https://www.genomicsengland.co.uk/research/academic/join-gecip. GS data may be accessed through an application process described here: https://www.ed.ac.uk/generation-scotland/for-researchers/access. The summary statistics data generated in this study are provided in the Supplementary Information file. The GWAS data used in this study are available in the GWAS catalog database under accession codes listed in Supplementary Data 10. The PGS scores generated in this study are available in the PGS Catalog under publication ID: PGP000618 and score IDs: PGS004869-PGS004886. The GBD data are publicly available https://vizhub.healthdata.org/gbd-results/ and the files used in this study are also available via our GitHub repository[76] under AbsoluteRiskEstimation. All other data generated during this study are included in this published article and its supplementary information files.

## Code availability

The code used for these analyses is available at https://github.com/intervene-EU-H2020/flagship [76].

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

## Acknowledgements

We want to acknowledge the participants and investigators of the FinnGen study. The FinnGen project is funded by two grants from Business Finland (HUS 4685/31/2016 and UH 4386/31/2016) and the following industry partners: AbbVie Inc., AstraZeneca UK Ltd, Biogen MA Inc., Bristol Myers Squibb (and Celgene Corporation & Celgene International II Sàrl), Genentech Inc., Merck Sharp & Dohme LCC, Pfizer Inc., GlaxoSmithKline Intellectual Property Development Ltd., Sanofi US Services Inc., Maze Therapeutics Inc., Janssen Biotech Inc, Novartis AG, and Boehringer Ingelheim International GmbH. Following biobanks are acknowledged for delivering biobank samples to FinnGen: Auria Biobank (www.auria.fi/biopankki), THL Biobank (www.thl.fi/biobank), Helsinki Biobank (www.helsinginbiopankki.fi), Biobank Borealis of Northern Finland (https://www.ppshp.fi/Tutkimus-ja-opetus/Biopankki/Pages/Biobank-Borealis-briefly-in-English.aspx), Finnish Clinical Biobank Tampere (www.tays.fi/en-US/Research_and_development/Finnish_Clinical_Biobank_Tampere), Biobank of Eastern Finland (www.ita-suomenbiopankki.fi/en), Central Finland Biobank (www.ksshp.fi/fi-FI/Potilaalle/Biopankki), Finnish Red Cross Blood Service Biobank (www.veripalvelu.fi/verenluovutus/biopankkitoiminta), Terveystalo Biobank (www.terveystalo.com/fi/Yritystietoa/Terveystalo-Biopankki/Biopankki/) and Arctic Biobank (https://www.oulu.fi/en/university/faculties-and-units/faculty-medicine/northern-finland-birth-cohorts-and-arctic-biobank). All Finnish Biobanks are members of BBMRI.fi infrastructure (www.bbmri.fi). Finnish Biobank Cooperative -FINBB (https://finbb.fi/) is the coordinator of BBMRI-ERIC operations in Finland. The Finnish Biobank data can be accessed through the Fingenious® services (https://site.fingenious.fi/en/) managed by FINBB. GS received core support from the Chief Scientist Office of the Scottish Government Health Directorates [CZD/16/6] and the Scottish Funding Council [HR03006] and is currently supported by the Wellcome Trust [216767/Z/19/Z]. Genotyping of the GS:SFHS samples was carried out by the Genetics Core Laboratory at the Edinburgh Clinical Research Facility, University of Edinburgh, Scotland and was funded by the Medical Research Council UK and the Wellcome Trust (Wellcome Trust Strategic Award "STratifying Resilience and Depression Longitudinally" (STRADL) Reference 104036/Z/14/Z). We thank participants and scientists involved in making the UK Biobank resource available (http://www.ukbiobank.ac.uk/). The Massachusetts General Brigham Biobank is described further here: https://www.massgeneralbrigham.org/en/research-and-innovation/participate-in-research/biobank. The Trøndelag Health Study (HUNT) is a collaboration between HUNT Research Center (Faculty of Medicine and Health Sciences, Norwegian University of Science and Technology NTNU), Trøndelag County Council, Central Norway Regional Health Authority, and the Norwegian Institute of Public Health. EstBB was funded by the European Union through the European Regional Development Fund Project no. 2014-2020.4.01.15-0012 GENTRANSMED. Data analysis was carried out in part in the High-Performance Computing Center of the University of Tartu. The EstBB research team received funding from the Estonian Research Council grant TT17 "Estonian Centre for Genomics". This research was made possible through access to the data and findings generated by the 100,000 Genomes Project. The 100,000 Genomes Project is managed by Genomics England Limited (a wholly-owned company of the Department of Health and Social Care). The 100,000 Genomes Project is funded by the National Institute for Health Research and NHS England. The Wellcome Trust, Cancer Research UK and the Medical Research Council have also funded research infrastructure. The 100,000 Genomes Project uses data provided by patients and collected by the National Health Service (NHS) as part of their care and support. We acknowledge the contribution of the Genomics England Research Consortium to the 100,000 Genomes Project. The complete list of members of this Consortium can be found in the Supplementary Materials. We acknowledge Pedro Gomes, Fiona Hagenbeek and Benjamin Wingfield of INTERVENE for help in submitting the PGSs to the PGS catalog. This project has received funding from the European Union's Horizon 2020 research and innovation programme under grant agreement No 101016775. N.M. is the recipient of funding by the Academy of Finland (grant numbers 331671 and 355567), the University of Helsinki HiLIFE Fellows Grant 2023-2025 and Finska Läkaresällskapet. K.L. and R.M. received funding from the Estonian Research Council grant PUT (PRG1911) and the Estonian Research Council grant TK (TK214). This work was supported by the German Research Foundation (DFG, grant number 516649954 to H.O.H.).

## Author contributions

S.R. and A.G. conceived and designed the study and supervised this work. B.J., K.L., and B.N.W. led the meta-analysis, contributed to study design and planning, and wrote the manuscript with comments from all co-authors. B.J., K.L., B.N.W., Y.W., K.Z., Y.C., M.K., and S.K. performed statistical and computational analyses. Z.Y., T.H., R. Monti, J.W., O.Y., C.L., D.H., Y.O., D.L.M., C.H., R. Marioni, S.F., A.R., A.R.M., B.M.N., K.H., R. Magi, A.P., H.H., N.M., A.G., and S.R. provided critical inputs to the interpretation of the data and approved the final version of the manuscript.

## Competing interests

Kristi Läll has participated as an analyst in a collaboration research project at the Institute of Genomics, University of Tartu, which was funded by Geneto OÜ. Andrea Ganna is the founder of Real World Genetics Oy. Bradley Jermy became an employee of BioMarin after this work was completed. No other authors have competing interests to declare.

## Additional information

[1]Institute for Molecular Medicine Finland, FIMM, HiLIFE, University of Helsinki, Helsinki, Finland. [2]Estonian Genome Centre, Institute of Genomics, University of Tartu, Tartu, Estonia. [3]K. G. Jebsen Center for Genetic Epidemiology, Department of Public Health and Nursing, Faculty of Medicine and Health, Norwegian University of Science and Technology, Trondheim, Norway. [4]Analytic and Translational Genetics Unit, Massachusetts General Hospital, Boston, MA, USA. [5]Stanley Center for Psychiatric Research and Program in Medical and Population Genetics, Broad Institute of MIT and Harvard, Cambridge, MA, USA. [6]Med Biotech Hub and Competence Center, Department of Medical Biotechnologies, University of Siena, Siena, Italy. [7]Centre for Genomic and Experimental Medicine, Institute of Genetics and Cancer, University of Edinburgh, Edinburgh, UK. [8]William Harvey Research Institute, Barts and the London School of Medicine and Dentistry, Queen Mary University of London, London, UK. [9]Hasso Plattner Institute, Digital Engineering Faculty, University of Potsdam, Potsdam, Germany. [10]Helsinki Biobank, Hospital District of Helsinki and Uusimaa (HUS), Helsinki, Finland. [11]Pathology Department, University of Helsinki, Helsinki, Finland. [12]Hasso Plattner Institute for Digital Health, Icahn School of Medicine at Mount Sinai, New York, NY, USA. [13]Blizard Institute, Barts and the London School of Medicine and Dentistry, Queen Mary University of London, London, UK. [14]Department of Genome Informatics, Graduate School of Medicine, the University of Tokyo, Tokyo, Japan. [15]Department of Statistical Genetics, Osaka University Graduate School of Medicine, Suita, Japan. [16]Laboratory for Systems Genetics, RIKEN Center for Integrative Medical Sciences, Kanagawa, Japan. [17]Laboratory of Statistical Immunology, Immunology Frontier Research Center (WPI-IFReC), Osaka University, Suita, Japan. [18]Medical Research Council Human Genetics Unit, Institute of Genetics and Cancer, University of Edinburgh, Edinburgh, UK. [19]Department of Electrical, Electronic and Information Engineering "Guglielmo Marconi", University of Bologna, Bologna, Italy. [20]Medical Genetics, University of Siena, Siena, Italy. [21]Genetica Medica, Azienda Ospedaliera Universitaria Senese, Siena, Italy. [22]Levanger Hospital, Nord-Trøndelag Hospital Trust, Levanger, Norway. [23]Massachusetts General Hospital, Broad Institute of MIT and Harvard, Cambridge, MA, USA. [24]Department of Public Health, University of Helsinki, Helsinki, Finland. [25]These authors contributed equally: Bradley Jermy, Kristi Läll, Brooke N. Wolford. [26]These authors jointly supervised this work: Andrea Ganna, Samuli Ripatti. ✉e-mail: andrea.ganna@helsinki.fi; samuli.ripatti@helsinki.fi

## Estonian Biobank research team

Reedik Mägi [ID] [2]

## FinnGen

Aarno Palotie [ID] [1,4,5], Henrike Heyne [ID] [1,9,12], Nina Mars [ID] [1,4,5], Andrea Ganna[1,23,26] ✉ & Samuli Ripatti [ID] [1,23,24,26] ✉

