## [Peer Review File · Nature Communications]

A unified framework for estimating country-specific cumulative incidence for 18 diseases stratified by polygenic riskReviewer #1 (Remarks to the Author):

Through work with the INTERVENE consortium, the authors present cumulative incidence from birth to age 80, specific to age, sex, and country, and stratified by PGS for 18 diseases, based on data from large biobanks. They use external data from large published GWAS to estimate PGS, and registry data from Global Burden of Disease for incidence. They also present an analytical framework to estimate cumulative incidence in biobanks, with available data on GWAS and age-, sex-, and country-specific incidence data. They also present how their estimates could have impact on clinical screening, and importantly show that PGS has influence on cumulative incidence that was age and sex dependent. This work is potentially significant, and is rigorous with thorough analytical methods, careful analyses that include sensitivity analyses. However, there is a lack of discussion of relevance of this work to gene-environment interactions, which is a missed opportunity. The article could be improved with consideration of the specific points outlined below.

Major comments

- A strength of the study is that it required data from the biobanks and GWAS samples to be independent. Was there any training and testing for the PGS? Did the authors consider using prior published PGS? Many of these traits have published scores (e.g. Mavadatt et al, AJHG for breast cancer).
- It is not clear which phenotypes were assessed for which studies—were all phenotypes assessed for all studies?
- How follow-up time is considered is unclear. How was this defined for each study? Were ICD codes used at any timepoint? Will this miss older cases, and favor younger cases? It would be helpful to see more detail for the distribution of the follow-up time by study. It seems that Table 1 may have some discrepancies, particularly for UKB—is there really 24 years of follow-up for everyone? Many studies have IQR=0, which seems odd.
- What about other ancestries? Framework for country specific estimates likely will not be applicable for non-Western European/non-White populations. It would be helpful to elaborate more on ancestry in the limitations paragraph beyond a single sentence.
- Including age and sex interactions with PGS in the survival analysis models is a strength. However, it is important to note that the PGS do not include age or sex interactions with the genetic variants. Does your work suggest that PGS should be developed separately by age and sex? Also, when testing for interaction effect, how were the meta-analyses performed? This is not straightforward with meta-analysis, since the interpretation of the interaction effect depends on the main effects.
- Lack of discussion of gene-environment (G-E) interactions is a missed opportunity. Many publications have cited lack of interactions between PGS scores and environmental factors as evidence that there are no G-E effects; but this work suggests that PGS score performs differently in different groups, such as by age and sex, and this may extend more broadly. In fact, can the findings that the impact of PGS on disease risk changes with age be interpreted as evidence of G-E?
- Results of Figure 2 are interesting, examining the heterogeneity of PGS effects across sex and age. How much are the trends influenced by GWAS sumstats? These are typically pooled across age and sex, so don't account for G-E. Authors note that differences in cumulative risk are driven by differences in baseline risk—isn't this mostly by design? They allowed baseline rates to differ, but the PGS itself does not differ. This may suggest the need to consider G-E in the development of PGS.
- I like the analyses that explore potential screening recommendations based on PGS. The large differences in ages to start screening illustrates the potential benefits of tailored screening; differences in age to start screening go away when we do not take into account the age and sex specific effects. Can we interpret this as evidence that we don't see GE if we aren't properly looking for it? I also noted that the bias in estimates are in the tails of the distribution (as expected); what are the implications of this for tailored screening, when we care most about the tails? Maybe this should be mentioned in the discussion.
- The paragraph about sex differences and the need for sex-specific GWAS is particularly important. Below are a couple references you may want to consider related to this, that are not limited to cardiovascular traits. Khramtsova et al, Cell, 2023, "Quality control and analytic best practices for testing genetic models of sex differences in large populations", PMID: 37172561; Khramtsova, Davis,

Stranger, Nat Rev Gen, 2019, "The role of sex in the genomics of human complex traits"; PMID: 30581192.

Minor comments

- The subscript $k+5$ in Equation 2 does not seem to be defined
- Notation is unclear in Equation 5. What is n_o ?
- What is meant by 'median age at onset for that interval'? Page 9, line 429
- Figure 1: legend should clearly state what the HR's are for (i.e. effect PGS on time to disease onset)
- First paragraph of results: doesn't discuss UKB, isn't that cohort the oldest?
- For sensitivity analyses with UKB, it is unclear why only some phenotypes were used, and it seems to be inconsistently described (four in line 693, six in line 700)

Reviewer #2 (Remarks to the Author):

Firstly, I am very impressed by the scope of this article. This is a highly ambitious work by the authors, incorporating a very large amount of data across multiple cohorts. Additionally, the authors demonstrate an excellent level of detail and awareness of the field in question. I believe that the methods shown in this paper are highly informative for the problem at hand (incorporating genetic information into epidemiology). I have a few questions and suggestions however, which would clarify the methods, which is the primary purpose of the paper as I read it.

1) The first paragraph of the introduction is only two sentences long, one of which just gives a list of examples of clinical calculators. It would be beneficial here to give more explanation of what clinical calculators are, how they are used, why they are so often used (e.g., simplicity, generalisability), and so on. Without this foundation, it is not clear to a reader why PGS would be so great.

2) In the methods the authors state that the PGS "were standardized to have mean 0 and variance 1" on a per dataset basis. Did they look at whether unstandardized or whole-dataset standardised PGS differed between studies? E.g., an older cohort such as UKBB might show fewer high-risk PGS individuals for breast cancer due to early mortality, than Genomics England which recruits younger individuals who are less likely to have experienced the disease yet.

3) Individuals were grouped by PGS strata into <20%, 20-40%, 40-60%, 60-80%, 80-90%, 90-95%, and >95%. Can the authors explain why these groupings were chosen, as the spacing appears to be arbitrary? Would it not make more sense to group by deciles, quintiles, or SD?

4) At the end of the "cumulative incidence estimation" section, the authors state that "ages outside of the range of the four HRs were assumed constant to the HR closest in age". For example, in Supplemental Figure 1, this would imply that a 10-year-old carries the same risk of T2D development as a 45-year-old. Perhaps it would be more parsimonious to simply not estimate age-specific hazard ratios for ages outside the ranges, or at least to limit them to more sensible numbers (e.g., +/- 10 years).

5) The authors could comment more in their discussion about the generalisability of population-specific PGSs to other populations (e.g., white European to east Asian) or to under-represented populations (e.g., North Africans), and how their results inform or are affected by this issue.

6) I don't believe the authors compare the performance of the different models (total, sex, age, age and sex, PGS strata) anywhere. It would strengthen their argument that this method is a good tool if they showed some measure of performance, e.g., explained variance from the predictors.

7) Supplementary Figures 2a-2e all have different x-axis limits for each disease. As such it is hard to immediately determine what diseases are higher or lower risk. Additionally, there is no clear marker

for many diseases of $HR=1$, so it is also not clear if risk is greater or lower than one. Please plot all figures on the same x-axis limits (e.g., 0.5 - 3.0) and clearly mark $HR=1$ with a vertical solid line or similar. An exception for, e.g., 2e melanoma would be fine as this goes up to $HR\sim=7$, so long as this is noted in the figure captions.

8) Supplementary Figure 3 is missing x-axis numbers for age.

Reviewer #1 (Remarks to the Author):

Through work with the INTERVENE consortium, the authors present cumulative incidence from birth to age 80, specific to age, sex, and country, and stratified by PGS for 18 diseases, based on data from large biobanks. They use external data from large published GWAS to estimate PGS, and registry data from Global Burden of Disease for incidence. They also present an analytical framework to estimate cumulative incidence in biobanks, with available data on GWAS and age-, sex-, and country-specific incidence data. They also present how their estimates could have impact on clinical screening, and importantly show that PGS has influence on cumulative incidence that was age and sex dependent. This work is potentially significant, and is rigorous with thorough analytical methods, careful analyses that include sensitivity analyses. However, there is a lack of discussion of relevance of this work to gene-environment interactions, which is a missed opportunity. The article could be improved with consideration of the specific points outlined below.

Major comments

- A strength of the study is that it required data from the biobanks and GWAS samples to be independent. Was there any training and testing for the PGS? Did the authors consider using prior published PGS? Many of these traits have published scores (e.g. Mavadatt et al, AJHG for breast cancer).

We thank the reviewer for this question. It has been shown that MegaPRS, incorporating several different Bayesian methods simultaneously, tends to outperform other methods (PMID: 34234142) and was therefore selected as a default method to generate all scores. We aimed for the most harmonized analysis possible across all traits, by selecting the same method for all and limiting our analysis to the single nucleotide polymorphisms (SNPs) in the intersection of HapMap phase 3 SNPs and the 1000 Genomes with a minor allele frequency greater than 1% in at least one super population (M=1,330,820). Therefore the scores across cohorts are including the same or very largely overlapping set of SNPs (please see Table 1) making the effect sizes of scores comparable across cohorts from the same disease. We've provided a table below from several cohorts showing the % overlap of SNPs within each trait's PGS.

To answer the question about training and testing of PGSs, MegaPRS by default generates from the full provided GWAS summary statistics two different sets called pseudo GWAS "training" and "testing" sets. The primary use of pseudo summary statistics is to construct and train prediction models, in order to decide parameters of the effect size's prior distribution (PMID: 34234142). Therefore no individual level data was used to train or test different versions of PGSs, however MegaPRS does internally

compose several versions of PGSs together with some statistics about their predictive ability.

We did consider using prior published PGS, but preferred to hold the PGS methodology constant across traits and cohorts. For example, breast cancer seems to have a slightly less polygenic architecture as PGS by the pruning and thresholding method perform better than a Bayesian method such as LDpred (PMID: 30104762). Whereas previous work by some of our co-authors has shown genome-wide scores are generally better than smaller scores (PMID: 35591975). Rather than wade into the score differences across traits, we selected a method that should perform broadly well (as shown in Zhang et al [PMID 34234142]). Future work would benefit from comparing existing scores, particularly ones being moved into clinical care, within this framework. As a comparison, the HR for the top 1% PRS versus the median (40-60%) in UK Biobank European individuals was 3.82 (3.46-4.21 95% CI) with the MegaPRS score in this study. In another publication using UK Biobank, the same comparison had an HR of 3.52 (2.93-4.24) using the Mavaddat, AJHG, 2019 score (PMID: 32596635)

Table 1. The percentage of SNPs in the PGS weights file that are present and included in each cohort by trait.

Phenotype	Genomics England	FinnGen	HUNT	MGBB	EstBB
Breast Cancer	99.70%	97.29%	100%	83.9%	98.2%
Epilepsy	99.70%	98.06%	100%	89.2%	98.6%
Gout	99.70%	96.94%	100%	85.7%	98.0%
Prostate Cancer	99.70%	96.98%	100%	83.4%	98.2%
Rheumatoid Arthritis	99.70%	96.88%	100%	88%	98.3%
Type 1 Diabetes	99.70%	90.70%	100%	95.7%	97.8%
All Cancers	99.70%	97.06%	100%	83.5%	98.3%
Atrial	99.70%	97.10%	100%	94.7%	98.3%

Fibrillation					
Coronary Heart Disease	99.70%	97.52%	100%	88.6%	98.3%
Type 2 Diabetes	99.70%	97.26%	100%	84.1%	98.4%
Skin Melanoma	99.70%	97.09%	100%	84.9%	98.3%
Asthma	99.70%	97.36%	100%	84.9%	98.4%
Major Depression	99.80%	98.21%	100%	86.9%	98.8%
Lung Cancer	99.80%	97.16%	100%	84.9%	98.4%
Hip Osteoarthritis	99.70%	97.46%	100%	84.9%	98.5%
Knee Osteoarthritis	99.70%	97.21%	100%	84.9%	98.4%
Appendicitis	99.70%	97.24%	100%	86.5%	98.5%
Colorectal Cancer	99.70%	97.27%	100%	83.6%	98.4%

• It is not clear which phenotypes were assessed for which studies—were all phenotypes assessed for all studies?

All phenotypes were not assessed for all studies due to either insufficient case numbers, poor phenotype definition within a specific study, or non-independent PGS (e.g. the study was included in the GWAS used to make the PGS in such a large quantity that could not be excluded for PGS analysis). In Supplementary Table 7 we have described which phenotypes were analyzed in each biobank together with some other descriptive statistics. We have also added a sentence (page 10, line 32) describing how many phenotypes were available for PGS-disease modeling in each biobank. For convenience, the numbers are also presented here in Table 2:

Table 2. Number of diseases available for analysis per biobank. Only 6 traits were analysed in UK Biobank due to overlapping samples with the GWAS used to build the PGS.

Biobank	Number of diseases analyzed
Estonian Biobank	18
FinnGen	18
Generation Scotland	15
Genomics England	18
HUNT	16
Mass General Brigham	17
UK Biobank	6

- How follow-up time is considered is unclear. How was this defined for each study? Were ICD codes used at any timepoint? Will this miss older cases, and favor younger cases? It would be helpful to see more detail for the distribution of the follow-up time by study. It seems that Table 1 may have some discrepancies, particularly for UKB—is there really 24 years of follow-up for everyone? Many studies have IQR=0, which seems odd.

We thank the reviewer for this question. In Table 1 in manuscript, IQR=0 for studies where follow-up for everyone has been defined as the start of registry data until the end of last linking between the registry and the biobank. We have now clarified how follow-up time is calculated differently across biobanks in the legend of Table 1. Detailed information about time coverage of ICD-10 codes and follow-up time calculation for each biobank is given in Supplemental Methods.

For some biobanks like FinnGen, registry follow-up is very long, covering the majority of individuals from birth. FinnGen also has different follow-up times depending on which registry is used for endpoint definition. For some biobanks, e.g., Estonian Biobank, nation-wide coverage for the majority of diagnosis starts from 2003 (except for cancers, which is covered from 1960s). Information about diseases present before 2003 are covered by self-reported information, indicating that some older cases indeed might

be omitted from data. In another manuscript by some of our co-authors (PMID: 32273609) a sensitivity analysis was conducted using full follow up for FinnGen without excluding cases that are prevalent with respect to the study sampling time (Figure 1) versus the same analysis in FINRISK excluding cases prevalent at the study start (Extended Data Figure 6). The survival curves were very similar.

In this analysis for the Cox regression models, follow-up time was always defined since birth. Meaning, in the survival models, left truncation was not accounted for, making the timescale in Cox proportional models to be age from birth. This is described in Methods 'Survival Analysis Models.' In Supplementary Figure 14 we show the impact of assuming follow-up begins at birth, at the start of the registry linkage or at recruitment in the UK Biobank has negligible effect on Hazard Ratio estimates.

- What about other ancestries? Framework for country specific estimates likely will not be applicable for non-Western European/non-White populations. It would be helpful to elaborate more on ancestry in the limitations paragraph beyond a single sentence. We thank the reviewer for this important point. We are currently working on a multi-ancestry version of this project. We have added an additional sentence on ancestry limitations (line 2, page 15).

- Including age and sex interactions with PGS in the survival analysis models is a strength. However, it is important to note that the PGS do not include age or sex interactions with the genetic variants. Does your work suggest that PGS should be developed separately by age and sex?

We do think that, in theory, for some traits the PGS should be developed separately by age and sex. It is possible that different set of SNPs should be used to compose sex- and -age specific scores together with age varying effect sizes like has been shown for menopause (PMID: 37543033). However, we would likely need a massive sample size to get precise genetic effect estimates for reference populations, as dividing current samples into age and sex specific groups would decrease the number of both cases and controls in each subGWAS and increase the standard errors of SNP effect sizes, making the accurate estimation of PGS more difficult. Nevertheless, this idea has been applied in practice (age-specific GWAS) for type 2 diabetes and it has been shown to reveal subgroup specific genetics (PMID: 33972266). As age-and-sex specific GWASs are not currently available on a large scale to build more elaborate PGS or their sample size is considerably smaller than overall GWASs, we believe we've made the optimal compromise to model the age and sex on the backend of the prediction, as we have done. This approach has been done before for breast cancer (PMID: 30554720).

Also, when testing for interaction effect, how were the meta-analyses performed? This is not straightforward with meta-analysis, since the interpretation of the interaction effect depends on the main effects.

Thank you for bringing this up. Meta-analysis was performed in a classical inverse *variance-weighted* average method. As PGS were standardised across cohorts, the betas for PGSs are for the same units across all cohorts. Then, when meta-analysing interaction effects, we are interested in if the interaction term was different from zero (i.e. if the difference of PGS betas for men and women were zero for instance). We believe that for that kind of analysis, classical meta-analysis approach was appropriate. However, we agree with the reviewer that meta-analyses of interaction effects are not straightforward given their dependence on the main effects. But the sex-stratified analysis provide the same conclusions, so we have more evidence for this approach.

- Lack of discussion of gene-environment (G-E) interactions is a missed opportunity. Many publications have cited lack of interactions between PGS scores and environmental factors as evidence that there are no G-E effects; but this work suggests that PGS score performs differently in different groups, such as by age and sex, and this may extend more broadly. In fact, can the findings that the impact of PGS on disease risk changes with age be interpreted as evidence of G-E?

We thank the reviewer for bringing up the importance of gene-environment interaction (GxE). Since this is difficult to formally test for in our setting, we do not feel justified making any strong statements about GxE in the manuscript. We are investigating whether or not the predictive ability of PGS differs across demographic variables, but age and sex are not considered classically environmental risk factors. The fact that risk prediction does differ across these non-genetic factors, does not necessarily translate into evidence of GxE. In Supplementary Figure 2 we show there is even heterogeneity of PGS effects across biobanks, but this may be explained by several factors, including differential ascertainment of the biobanks and it is therefore hard to say if these are due to environmental factors. We feel it is not warranted to speculate on the extension of these results to GxE more broadly, but have expanded upon this in the discussion (Page 14, Line 12).

From our results, we see downstream opportunities to look at differential PGS effects across other non-genetic factors. We are currently looking into PGS effects stratified by more classical environmental factors such as smoking and education which should help us say more about evidence of GxE. As a lot of attention goes into moving from relative scale to absolute scale estimates, this manuscript is a first step on using variables which are easily accessible and consistent across biobanks and can be easily extended on a country level. Even variables such as smoking are often not available in a unified way on a population-level for the use of this framework and therefore we

focused on age and sex which are almost always easily available for an individual on a country-level.

However, we expanded a sentence in the discussion with regards to GxE and the interpretation of disease risk changing across ages (Page 13, Line 34). We also cite Jiang et al (PMID: 34437535), a study which loosely links age-specific genetic effects to the environment. They believe age-specific effects are due to the accumulation of environmental factors over time. However, we cannot directly test for this in our framework.

- Results of Figure 2 are interesting, examining the heterogeneity of PGS effects across sex and age. How much are the trends influenced by GWAS sumstats?

We acknowledge that GWAS summary statistics depend on the makeup of the GWAS cohorts and are influenced by the proportion of females, advanced age cases, clinical vs population-based studies, etc. We've added a sentence to the discussion highlighting this (Page 15, Line 10).

These are typically pooled across age and sex, so don't account for G-E. Authors note that differences in cumulative risk are driven by differences in baseline risk—isn't this mostly by design? They allowed baseline rates to differ, but the PGS itself does not differ. This may suggest the need to consider G-E in the development of PGS.

We thank the reviewer for this interesting interpretation. We designed the study so that country-specific differences in baseline risk captured by GBD were included in the cumulative risk as it has been shown that biobank-based baseline risks are not as accurate due to participation bias (PMID 33888908). At the moment, the PGS are built on GWAS that are not age- and sex- stratified. We agree that for some diseases, Stratified GWASs would provide a more optimal base for PGS construction. We have revised a sentence in the discussion and now suggesting that future GWAS should provide summary statistics stratified by age and sex (Page 14, Line 7).

- I like the analyses that explore potential screening recommendations based on PGS. The large differences in ages to start screening illustrates the potential benefits of tailored screening; differences in age to start screening go away when we do not take into account the age and sex specific effects. Can we interpret this as evidence that we don't see GE if we aren't properly looking for it?

We thank the reviewer for this interesting question. For diseases where we don't see significant age- and sex-specific effects, it likely doesn't matter. For diseases where age and sex specific models are the best, when moving to absolute risk estimation (e.g., 10-year risk), it might be, that in older age, where PGS effects seem to be lower, the absolute risk estimate in high genetic risk group individuals might be similar to average

genetic risk group estimate in younger age group. We agree for the most accurate absolute risk estimation, the age- and sex-specific PGS effects should be incorporated.

I also noted that the bias in estimates are in the tails of the distribution (as expected); what are the implications of this for tailored screening, when we care most about the tails? Maybe this should be mentioned in the discussion.

We agree we need to take into account the uncertainty of the cumulative estimates in PGS tails. One option is to also focus on the values of the confidence intervals. For example one could say that the lower 95% confidence interval for the cumulative incidence has to exceed some pre-defined threshold. For instance for breast cancer (PMID: 37568754), study defined that high risk individuals are individuals, whose 10-year risk estimate is two times greater than the estimated average for a 50-year old from the general population. We also believe it matters how the intervention will be modified based on PGS group estimate - if it is more frequent screening, the variation of the estimates might not be that important compared to when interventions are more invasive (e.g. medication, surgery). We have added a sentence on this topic in the discussion (Page 14, Line 34).

- The paragraph about sex differences and the need for sex-specific GWAS is particularly important. Below are a couple references you may want to consider related to this, that are not limited to cardiovascular traits. Khramtsova et al, Cell, 2023, “Quality control and analytic best practices for testing genetic models of sex differences in large populations”, PMID: 37172561; Khramtsova, Davis, Stranger, Nat Rev Gen, 2019, “The role of sex in the genomics of human complex traits”; PMID: 30581192.

Thank you for these important references. We have reviewed them and added them (Page 13, Line 44). We’ve also added a sentence regarding a recent paper PMID: 37228747 exploring Gene X Sex which has evidence to suggest there are sex differences in the magnitude of genetic effects rather than in the actual causal variants (Page 13, Line 34).

Minor comments

- The subscript $k+5$ in Equation 2 does not seem to be defined
We’ve made the k subscript’s definition more clear in the text directly above the equation. We have also made changes in the subscript. $[m, m+4]$ were used in the NEJM article because ages were considered only in discrete form (ie 25-29, 30-34). We use $[m, m+5]$ to be mathematically more precise as exponential distribution is continuous.
- Notation is unclear in Equation 5. What is n_o ?

We've now defined this below the equation and adjusted our terminology from baseline to reference group.

- What is meant by 'median age at onset for that interval'? Page 9, line 429

We've now better defined this in the text and cited the Supplementary Figure 1.

- Figure 1: legend should clearly state what the HR's are for (i.e. effect PGS on time to disease onset)

We've added this to the legend.

- First paragraph of results: doesn't discuss UKB, isn't that cohort the oldest?

All biobanks are discussed more in detail in the supplement, however UKB is not the oldest cohort.

- For sensitivity analyses with UKB, it is unclear why only some phenotypes were used, and it seems to be inconsistently described (four in line 693, six in line 700)

Of the 6 traits we considered for sensitivity analysis, we only consider 4 to be most influenced by primary care definitions. We didn't evaluate use of primary care data for breast and prostate cancer because these are generally tagged by hospital diagnoses and cancer registries. We have made this clear in the Sensitivity Analyses results section.

Reviewer #2 (Remarks to the Author):

Firstly, I am very impressed by the scope of this article. This is a highly ambitious work by the authors, incorporating a very large amount of data across multiple cohorts. Additionally, the authors demonstrate an excellent level of detail and awareness of the field in question. I believe that the methods shown in this paper are highly informative for the problem at hand (incorporating genetic information into epidemiology). I have a few questions and suggestions however, which would clarify the methods, which is the primary purpose of the paper as I read it.

1) The first paragraph of the introduction is only two sentences long, one of which just gives a list of examples of clinical calculators. It would be beneficial here to give more explanation of what clinical calculators are, how they are used, why they are so often used (e.g., simplicity, generalisability), and so on. Without this foundation, it is not clear to a reader why PGS would be so great.

Thank you for this suggestion. We've now incorporated more context about risk calculators to the introductory paragraph (Page 3, Line 7).

2) In the methods the authors state that the PGS "were standardized to have mean 0 and variance 1" on a per dataset basis. Did they look at whether unstandardized or whole-dataset standardised PGS differed between studies? E.g., an older cohort such as UKBB might show fewer high-risk PGS individuals for breast cancer due to early

mortality, than Genomics England which recruits younger individuals who are less likely to have experienced the disease yet.

We thank the reviewer for this point and agree. We do not require a harmonized set of SNPs (see table on page 3 of this document) across cohorts so direct comparison of means of the scores across biobanks would not be fair as the set of SNPs in each biobank is not exactly the same. However, we agree an older cohort may have fewer high risk individuals due to mortality. In a perfect world, you'd have a true population mean and standard deviation of a PGS. Currently, we only have a subset of adult individuals, who are capable of joining biobanks. This means that distribution characteristics are likely to be under-estimated in biobank settings for diseases with highest mortality in early age (e.g., myocardial infarction or early onset breast cancer). While this may be present, the PGS estimates are combined GBD estimates to calculate lifetime risks, so we hope that this corrects for the biobank participation bias. We've added a sentence to the discussion outlining this limitation (Page 15, Line 5).

To look into this matter more deeply, below we show the means and standard deviations of raw PGS for selected traits in UKBB and Genomics England. We don't see smaller means in Genomics England except for T1D and Epilepsy. Although UKB has a median follow up time of 24 years and Genomics England of 29 years. Differences in mean scores are more likely due to the recruitment strategy, with UKB being more population based and Genomics England is enriched for rare diseases and cancer.

Phenotype	UKBB EUR		Genomics England	
	Mean	SD	Mean	SD
GOUT	-0.1353564	0.28319884	-0.120283	0.297277
C3_PROSTATE	0.27176723	0.43055584	0.196047	0.441331
RHEUMA_SEROPOS_OTH	0.21610498	0.17326326	0.215894	0.235893
T1D	0.04753751	0.50829117	-0.0246306	0.914314
G6_EPLEPSY	-2.5158789	0.441555	-7.09081	0.680181
C3_BREAST	0.17783048	0.42791479	0.129746	0.519281

3) Individuals were grouped by PGS strata into <20%, 20-40%, 40-60%, 60-80%, 80-90%, 90-95%, and >95%. Can the authors explain why these groupings were chosen, as the spacing appears to be arbitrary? Would it not make more sense to group by deciles, quintiles, or SD?

Thank you for this question. It is our understanding that groupings of PGSs are often arbitrary and mainly done for illustrative purposes (cumulative incidence profiles in

different groups) or to find PGS groups which have equivalent risk to monogenic mutations (<https://www.ncbi.nlm.nih.gov/pmc/articles/PMC6128408/>). We have tried to present PGS results (HR) in a diverse way—separately for each biobank for 1 SD in Supplementary Figure 2a and 2b and per 1 SD and per quartiles of PGS in meta-analysed form in Supplementary Table 8.

We also based our PGS quartile groupings of Mavaddat et al, Figure 3 groupings (<https://pubmed.ncbi.nlm.nih.gov/30554720/>). We had higher resolution at the top end of the PGS distribution where screening has the most potential (e.g., >99%), but we were often underpowered to have symmetric resolution at the bottom end of the distribution (e.g. <1%) when stratifying by age and sex.

4) At the end of the “cumulative incidence estimation” section, the authors state that “ages outside of the range of the four HRs were assumed constant to the HR closest in age”. For example, in Supplemental Figure 1, this would imply that a 10-year-old carries the same risk of T2D development as a 45-year-old. Perhaps it would be more parsimonious to simply not estimate age-specific hazard ratios for ages outside the ranges, or at least to limit them to more sensible numbers (e.g., +/- 10 years).

We’ve clarified the wording from intervals to quartiles (page 8, line 9). Because we need to estimate hazard ratios across the lifetime we must estimate age-specific hazard ratios for ages outside the quartiles. We decided to assume the risk is equal for some individuals because we are generally underpowered with relatively few cases below middle age and censoring above age 80. If we could estimate HRs between age 20-40 we believe this would be higher than the HR we are using, and between 70-80 it would be lower than what we are using, so we believe ours to be a more conservative approach. In the future with larger data sets, a more fine-scale age grouping, such as deciles, would be ideal. We do allow the baseline hazards to differ across the lifetime.

5) The authors could comment more in their discussion about the generalisability of population-specific PGSs to other populations (e.g., white European to east Asian) or to under-represented populations (e.g., North Africans), and how their results inform or are affected by this issue.

We thank the reviewer for this suggestion. We have added additional text regarding ancestry in the discussion (Page 15, Line 2).

6) I don’t believe the authors compare the performance of the different models (total, sex, age, age and sex, PGS strata) anywhere. It would strengthen their argument that this method is a good tool if they showed some measure of performance, e.g., explained variance from the predictors.

We thank the reviewer for this suggestion. Unfortunately, evaluating model performance with standard metrics such as Harrell's C-statistic is not entirely straightforward for all of our models due to stratification by PGS and sex and age as the time scale. Below we've included manuscript text outlining model selection criteria.

a) Sex-specific effects were chosen if:

1. The meta-analyzed interaction effect (PGS*Sex) was significant ($p < 2.8 \times 10^{-3}$, details below)

b) Age-specific effects were chosen if:

1. There was a significant heterogeneity across the four quartiles, estimated using a Cochran's Q test (39).

c) Age and sex-specific effects were chosen if:

1. Separate age and sex specific effects were found in both tests a and b.
2. Age-specific effects were found in a single sex where not previously found.
3. Age-specific effects were found to differ significantly across sexes.

We estimate C-index in one of our cohorts, HUNT, for an example trait, T2D, which shows age and sex specific effects.

The first 10 genetic PCs are also used as covariates and as the only predictors in the model, the C-index is 0.52 (SE=5.6E-3).

For a) we have a C-index of 0.687 (SE=4.8E-3) with PGS percentile groupings and a C-index of 0.693 (SE=4.8E-3) when we add sex to this model. Similar results are seen if we use the PGS per SD as a continuous variable rather than the percentile grouping.

For b) if we just use the first 10 PCs and birth year as a surrogate for age, the C-index is 0.779 (SE=2.9E-3). When we add in the PGS strata groupings this becomes 0.824 (SE=2.9E-3).

For c) we use the first 10 PCs, birth year, and sex and the C-index is 0.783 (SE=2.9E-3). When we add the PGS strata groupings as the full model for T2D, this becomes 0.827 (SE=2.9E-3).

7) Supplementary Figures 2a-2e all have different x-axis limits for each disease. As such it is hard to immediately determine what diseases are higher or lower risk. Additionally, there is no clear marker for many diseases of HR=1, so it is also not clear if risk is greater or lower than one. Please plot all figures on the same x-axis limits (e.g., 0.5 - 3.0) and clearly mark HR=1 with a vertical solid line or similar. An exception for, e.g., 2e melanoma would be fine as this goes up to HR~7, so long as this is noted in the figure captions.

Thank you, we have replotted these figures as suggested.

8) Supplementary Figure 3 is missing x-axis numbers for age.
We have now corrected the x-axis labels.

Reviewer #1 (Remarks to the Author):

The authors have thoroughly addressed my comments and concerns. I have no further comments.

Reviewer #2 (Remarks to the Author):

Overall, I am satisfied with the responses to my and the other reviewers' comments. However, the text of the manuscript has not been altered in many instances. For example, my comment 3 (...PGS strata...). The authors make a coherent and well-thought out response, but there is no adaptation of the text to reflect this in the Methods or in the Discussion. If I have this question, doubtless some readers will also! Can the authors please check that they have responded to all comments and questions in the manuscript, not just in comments to reviewers.

My remaining concerns and comments are as follows:

1) One of the lead authors, Bradley Jermy, currently works at BioMarin Pharmaceutical Inc. as a Clinical Genomics Lead. While I do not believe this compromises the work produced here, this should be declared under Declarations of Interest. It would be advisable to recollect or confirm this information is still valid for all authors.

2) I like the paragraph that the authors have written for the start of the introduction. I am confused by the use of "often" in the second sentence (p3, line 8). If clinical calculators are only "often" weighted linear models, what are they the rest of the time? Please clarify/rewrite.

3) Regarding my question 4 in the first submission. I agree with the authors that determining exact hazard ratios outside of the age ranges for which they have data is difficult statistically and not meaningful clinically in many cases. I am wary of labelling this approach as "conservative" without context. E.g., an underestimation of prostate cancer PGS might lead to greater rates of later stage cancer diagnoses, which leads to poorer prognoses and greater medical expenditure. Conversely, an overestimation of gout PGS might lead to more dietary interventions and so reduce incidence and decrease medical expenditure. Whether any outcome is "conservative" here depends if you are a clinician, a patient, or an accountant. The authors should discuss the implications of this assumption in the Discussion for, e.g., incidence rates, intervention strategy, patient outcomes. I agree that with broader age data in future these estimates will become more accurate, and this should also be stated. Additionally, I do not understand what the last sentence of this response refers to ("We do allow the baseline hazards to differ across the lifetime").

4) Regarding my question 6 in the first submission. The authors respond that generating model performance statistics is "not straightforward". However, they then provide an example of C-index for Type 2 Diabetes in the HUNT cohort. The numbers they provide show a clear benefit of including age and sex effects into PGS strata. It would not be necessary to include C-indices for all possible configurations, as they have done in the response. A table of the simplest model and best model, per disease, per cohort, would be a very powerful demonstration. A simple superscript indicator with a footnote could show what the best model is (sex, age, or sex and age). Without any evaluation metrics, the reader is forced to take the authors' word that these models perform well. Describing these values is much more persuasive.

5) In the legend for Figure 2, "Group" should be clearer that it refers to PGS strata group.

6) Supplementary Figure 2 is still missing a reference line at HR=1. All other forest plots look much better now.

Reviewer #1 (Remarks to the Author):

The authors have thoroughly addressed my comments and concerns. I have no further comments.

We appreciated the opportunity to address your comments and concerns.

Reviewer #2 (Remarks to the Author):

Overall, I am satisfied with the responses to my and the other reviewers' comments. However, the text of the manuscript has not been altered in many instances. For example, my comment 3 (...PGS strata...). The authors make a coherent and well-thought out response, but there is no adaptation of the text to reflect this in the Methods or in the Discussion. If I have this question, doubtless some readers will also! Can the authors please check that they have responded to all comments and questions in the manuscript, not just in comments to reviewers.

Thank you for this suggestion. We originally hesitated to add additional text to the manuscript, but we have reviewed our previous responses for answers without in-text clarifications. We added to the text in the following places:

- Reviewer 2, point 3 now addressed in Methods (page 7)
- Reviewer 2, point 4 - please see #3 below
- Reviewer 2, point 6 - now addressed in Results (page 11) by citing the new Supplementary Figure 6
- Reviewer 1, point 1 - now addressed in a section of the Supplementary Material (page 8)
- Reviewer 1, point 5a - the possible need for age and sex stratified GWAS was previously added to the Discussion (top of page 14)
- Reviewer 1, point 5b: meta-analysis of interaction effects are now better described in Methods (page 6)
- Reviewer 1, point 8a - now addressed in Discussion (page 14)

My remaining concerns and comments are as follows:

1) One of the lead authors, Bradley Jermy, currently works at BioMarin Pharmaceutical Inc. as a Clinical Genomics Lead. While I do not believe this compromises the work produced here, this should be declared under Declarations of Interest. It would be advisable to recollect or confirm this information is still valid for all authors.

Thank you for reminding us of this. Co-author Bradley Jermy did not work at BioMarin at the time of this work, but we have made that clear in the Declarations of Interest. We have checked with all co-authors and added a new COI from Andrea Ganna. There are no additional declarations of interest from other co-authors. We also updated funding sources.

2) I like the paragraph that the authors have written for the start of the introduction. I am confused by the use of “often” in the second sentence (p3, line 8). If clinical calculators are only “often” weighted linear models, what are they the rest of the time? Please clarify/rewrite.

Thank you for the opportunity to clarify. We have revised the text in the introduction by removing the mention of weighted linear models, since this is not a particularly relevant detail for the point we are trying to make.

3) Regarding my question 4 in the first submission. I agree with the authors that determining exact hazard ratios outside of the age ranges for which they have data is difficult statistically and not meaningful clinically in many cases. I am wary of labelling this approach as “conservative” without context. E.g., an underestimation of prostate cancer PGS might lead to greater rates of later stage cancer diagnoses, which leads to poorer prognoses and greater medical expenditure. Conversely, an overestimation of gout PGS might lead to more dietary interventions and so reduce incidence and decrease medical expenditure. Whether any outcome is “conservative” here depends if you are a clinician, a patient, or an accountant. The authors should discuss the implications of this assumption in the Discussion for, e.g., incidence rates, intervention strategy, patient outcomes. I agree that with broader age data in future these estimates will become more accurate, and this should also be stated. Additionally, I do not understand what the last sentence of this response refers to (“We do allow the baseline hazards to differ across the lifetime”).

These are excellent examples of the context required to interpret whether an approach is conservative. We have expanded on this as a potential limitation in the discussion (page 15): “Finally, when modelling age-specific risk we used the hazard ratio estimated for the closest age quartile for ages less than the first age quartile and greater than the fourth age quartile (see Methods). Through this approach we did not estimate exact hazard ratios outside of the age ranges for which we have data and statistical power to do so, but our assumption of constant hazards outside the age quartiles would, for example, estimate a 20-year old carrying the same T2D risk as a 45 year old (Supplementary Figure 1). In a translational context, this could create underestimation or overestimation of risk leading to over or under treatment, unnecessary expenditures, missed diagnoses, etc. In the future with larger data sets, hazard ratios estimate on a higher resolution age grouping, such as deciles, would allow for more accurate estimates.”

In our response we said “We do allow the baseline hazards to differ across the lifetime.” What we meant was that in each 5-year age group for females and males separately, the hazard of a disease was estimated based on Global Burden of Disease estimates. Therefore the baseline hazard varies in by sex in age groups 0-5, 5-10, 10-15. This is described in the methods section “Cumulative incidence estimation.”

4) Regarding my question 6 in the first submission. The authors respond that generating model performance statistics is “not straightforward”. However, they then provide an example of C-index for Type 2 Diabetes in the HUNT cohort. The numbers they provide show a clear benefit of including age and sex effects into PGS strata. It would not be necessary to include C-indices for all possible configurations, as they have done in the response. A table of the simplest model and best model, per disease, per cohort, would be a very powerful demonstration. A simple superscript indicator with a footnote could show what the best model is (sex, age, or sex and age). Without any evaluation metrics, the reader is forced to take the authors’ word that these models perform well. Describing these values is much more persuasive.

Thank you for this suggestion. We have created a Supplementary Figure rather than Table, as it was easier to visualize the differences between C-statistic values. For example, it was obvious it was best to adjust for age and sex versus just age for Gout and CHD. Since some traits have a different “best” model for males versus females, we avoided notating which was defined as best, and pointed the reader to Supplementary Table 13 in the figure legend.

5) In the legend for Figure 2, “Group” should be clearer that it refers to PGS strata group.

We thank the reviewer for this suggestion and have made the change in the figure legend.

6) Supplementary Figure 2 is still missing a reference line at HR=1. All other forest plots look much better now.

We have updated Supplementary Figure 2e (page 15) to include the reference line at HR=1.

Reviewer #2 (Remarks to the Author):

I would like to thank the authors for the time and effort they have used in responding to my comments. I believe this is an excellent paper and I am glad to recommend its acceptance.